# Investigating Human-Identifiable Features Hidden in Adversarial Perturbations

## Abstract

Neural networks perform exceedingly well across various machine learning tasks but are not immune to adversarial perturbations. This vulnerability has implications for real-world applications. While much research has been conducted, the underlying reasons why neural networks fall prey to adversarial attacks are not yet fully understood. Central to our study, which explores up to five attack algorithms across three datasets, is the identification of human-identifiable features in adversarial perturbations. Additionally, we uncover two distinct effects manifesting within human-identifiable features. Specifically, the masking effect is prominent in untargeted attacks, while the generation effect is more common in targeted attacks. Using pixel-level annotations, we extract such features and demonstrate their ability to compromise target models. In addition, our findings indicate a notable extent of similarity in perturbations across different attack algorithms when averaged over multiple models. This work also provides insights into phenomena associated with adversarial perturbations, such as transferability and model interpretability. Our study contributes to a deeper understanding of the underlying mechanisms behind adversarial attacks and offers insights for the development of more resilient defense strategies for neural networks.

## 1 Introduction

Neural networks achieve an unprecedented level of performance across a vast array of machine learning tasks(Hinton et al., 2012) and many more applications are expected to emerge in the near future. Thus, it is particularly concerning that small changes to input data known as adversarial perturbations, which are often imperceptible to human eyes, can dramatically alter neural networks' judgments (Szegedy et al., 2014), thereby compromising their reliability. This vulnerability introduces significant risks to real-world applications of neural networks. For instance, an adversarial perturbation applied to a traffic sign could cause an autonomous car to misread a stop sign as a 45 mph speed limit sign. This misunderstanding could trigger a sudden acceleration, possibly resulting in accidents (Eykholt et al., 2018).

In this paper, we carefully examine the underlying properties of adversarial perturbations, which leads us to hypothesize that human-identifiable features exist within these perturbations. These features are often easily recognized by humans, such as the tire of a car, a cock's crest, or more generally, a particular object's shape. In the process of validating this hypothesis, we identify two factors obscuring human-identifiable features: excessive noise and incomplete feature information. To reveal the hidden features concealed by these two factors, both factors must be mitigated without altering the essence of the perturbations. Since different neural networks usually generate perturbations with noise and incomplete information different from each other, averaging these perturbations, derived from the same image, effectively minimizes the effects of noise. At the same time, assembling incomplete information leads to the emergence of human-identifiable features.

We find that, using the methodology described above, human-identifiable features emerge. In fact, two types of human-identifiable features show up in our study: the masking effect and the generation effect. The masking effect includes important features from the input image, potentially with a sign inversion. The generation effect adds new features to the original image, simulating a different class, which can potentially result in misclassification by both humans and neural networks.

We demonstrate our finding by using five different attack algorithms, including both gradient-based and search-based algorithms, and three different datasets, including MNIST (Deng, 2012), CIFAR-10 (Krizhevsky & Hinton, 2009), and ImageNet (Deng et al., 2009). We also quantify our results by evaluating the recognizability and attack strength of the perturbations. To further highlight that human-identifiable features are critical in causing the model to misclassify, we employ pixel-level annotations to extract these features from the perturbations. Our findings confirm that these features result in significantly more adverse attacks compared to other features, even if they correspond to a smaller attack surface and vector norm within such perturbations. Moreover, the "masking effect" results in an intriguing phenomenon: Different algorithms produce perturbations that, when summed across different independent models, converge to higher cosine similarity values.

Perturbations containing human-identifiable features enable us to explain important phenomena related to adversarial perturbations, including their transferability (Szegedy et al., 2014; Papernot et al., 2016), the enhancement of model explainability through adversarial training (Goodfellow et al., 2015; Tsipras et al., 2019; Ross & Doshi-Velez, 2018; Santurkar et al., 2019), and the role of non-robust features (Ilyas et al., 2019). It is noted that our experimental findings do not rule out other factors that could cause adversarial perturbations to impact the performance of neural networks. The key finding of the paper is summarized in Figure 1.

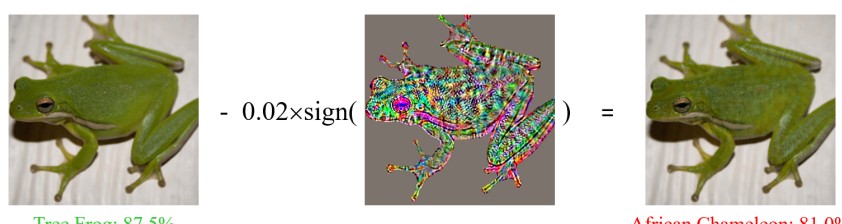

Tree Frog: 87.5%      African Chameleon: 81.0%

Figure 1: Overview of the key findings of this work. A DenseNet-121 model classifies the image of a tree frog correctly (87.5%). Averaging perturbations from multiple neural networks leads to the emergence of distinctive, frog-shaped features (middle panel). After subtracting the features from the original image, the DenseNet-121 model misclassifies the image as an African chameleon (81.0%).

## 2 RELATED WORK

Several works investigate the existence and/or the mechanisms behind adversarial perturbations. These include (1) the over-linearity in neural networks (Goodfellow et al., 2015; Tramèr et al., 2017), which indicates a linear relationship between input and output changes of a neural network. This allows small perturbations to impact output in high-dimensional input spaces. (2) The view that adversarial perturbations being non-robust features argues that neural networks, trained solely to minimize loss, may learn non-robust features irrelevant to human perception but predictive for model classification (Schmidt et al., 2018; Tsipras et al., 2019; Ilyas et al., 2019). Adversarial perturbations aim to alter these non-robust features, making them unrecognizable to humans yet influential in machine decisions. There are more works attempting to investigate this problem (Tanay & Griffin, 2016; Shafahi et al., 2019; Zhang et al., 2022; Han et al., 2023).

Elsayed et al. (2018) show that perturbations, generated under targeted attacks, using an ensemble of multiple models, and the inclusion of a retinal layer in each model decreases human classification accuracy of adversarial examples by approximately 10% in just 60-70 milliseconds. The authors suggest that a brief image display limits the brain's time for top-down processing, making the brain act like a feedforward neural network, hence the perturbations may fool humans. Athalye et al. (2017) and Brown et al. (2017) show that perturbations generated by targeted attack algorithms and augmented through the Expectation Over Transformation (EOT) technique increase their robustness in fooling neural networks and better align with human perception.

Differing from prior studies, we demonstrate the concept that human-identifiable features occur naturally in perturbations created by both untargeted and targeted attack algorithms, requiring no additional constraints. We further show that these features are effective in deceiving models and use this concept to elucidate several key phenomena.

## 3 ASSUMPTIONS MADE IN THIS STUDY

### 3.1 PERTURBATIONS CONTAIN HUMAN-IDENTIFIABLE FEATURES

In image classification, data is labeled by humans according to features that they can recognize. A well-trained model should rely, at least partially, on human-identifiable features to correctly classify the image. Hence, adversarial perturbations, which can fool models, will likely modify the features that the model's classification is based on. As a result, perturbations may incorporate human-identifiable features from the original image, which likely is a key factor that deceives models.

### 3.2 FACTORS CONCEALING HUMAN-IDENTIFIABLE FEATURES

In practice, human-identifiable features are often not readily visible in perturbations. We hypothesize that high noise levels and incomplete feature information in perturbations hinder the visibility of human-identifiable features. The origins of these two issues are explored below.

The neural network's gradient may be noisy, as indicated by Smilkov et al. (2017). Since adversarial perturbations are often derived from the gradient, they may also exhibit high noise levels.

Due to a model's limited computational capacity and challenges in achieving the global optimum of weight parameters during training, neural networks might not fully capture all useful information in the data for classification. Perturbations are specifically designed to maximize the model's loss function, thus affecting only the features the model has learned. As a result, such perturbations may contain only a subset of human-identifiable features. The extent and distribution of this incompleteness affect the visibility of these features.

## 4 EXPERIMENTAL METHOD

Our objective is to reduce noise and assemble incomplete features within adversarial perturbations, without changing their nature. Since the noise in perturbations originating from different models of the same image is likely independent, we in effect reduce the noise by averaging the perturbations generated by different neural networks. Because two perturbations from distinct models are unlikely to exhibit identical human-identifiable features, upon averaging, we in practice aggregate the incomplete components of the associated features. Additionally, we argue that averaging perturbations does not introduce any new information, as illustrated in Section 5.2.2, where the high attack success rate is maintained for averaged perturbations.

In order to overcome the two issues, it is necessary to acquire a sufficient number of neural networks to produce enough number of perturbations. However, the number of available neural networks is often limited. To solve this problem, we use a method inspired by SmoothGrad (Smilkov et al., 2017). In this method, we create multiple copies of a single input image, each incorporated with different Gaussian noise. This flexibility enables us to generate subtly different perturbations for each model using the same image, thereby increasing the sample size of perturbations. Such a process can potentially enhance the visibility of human-identifiable features through reducing the noise.

Mathematically, generating perturbations with multiple neural networks and the incorporation of Gaussian noise can be expressed as follows:

$$V(x) = \frac{1}{mn} \sum_{i=1}^{m} \sum_{j=1}^{n} V_i(x + N_{ij}(0, \sigma^2)) \tag{1}$$

Where $V$ denotes the generated perturbations, $x$ represents the input image, and $n$ and $m$ specify the number of image copies for each model and the total number of models used, respectively. Each perturbation $V_{ij}$ is generated from the $i^{th}$ model using the $j^{th}$ sample of Gaussian noise, represented by $N(0, \sigma^2)$, where the noise has a mean of 0 and a standard deviation of $\sigma$. Note that while adding Gaussian noise is useful, it is not essential for revealing human-identifiable features, which is further discussed in Appendix A.

### 4.1 EXPERIMENTAL SETUP

In the ImageNet experiment, we investigate perturbations generated by gradient-based attacks, including Basic Iterative Method (BIM) (Kurakin et al., 2016), CW attack (Carlini & Wagner, 2017), and DeepFool attack (Moosavi-Dezfooli et al., 2016) as well as search-based attacks, including Square attack (Andriushchenko et al., 2020), and One-pixel attack (Su et al., 2019). We will first discuss gradient-based attacks, followed by search-based attacks. For the experiment, we selected 20 classes[1] out of 1000 classes in the validation set and used the first 10 images in each class, yielding a total of 200 images. All images are scaled to a [0,1] pixel value range.

We divided all neural networks used in the experiment into two categories: source models and testing models. Source models are used to create adversarial perturbations. Testing models are used to evaluate both the attack strength and how obvious the human-identifiable features are in the perturbations.

Perturbations are generated in two distinct settings:

Single model setting (SM): In this setting, we generate perturbations in the usual way, i.e. sending an image into a single source model, specifically ResNet50 (He et al., 2016), and obtain corresponding perturbations, which serve as a baseline for comparison.

Multiple models with Gaussian noise setting (MM+G): In this setting, we generate perturbations according to Eqn. 1. We use Gaussian noise with a mean of 0 to generate perturbations. The variance is kept low to minimally affect perturbations and is determined by the employed attack algorithms. For the BIM attack, we use a standard deviation of 0.02. For both CW and DeepFool attacks, the standard deviation is 0.05. For each input image, we add 10 different Gaussian noise samples according to the above-mentioned method. We repeat this for each of the 270 source models used in the experiment, resulting in a total of 2,700 perturbations for each image. These 2,700 perturbations are then averaged to create one final perturbation for further analysis.

In our experiment, we download 274 models with diverse sets of architecture from PyTorchCV (Sémery, 2018). 270 models are used as source models in the MM+G setting, and the remaining four are designated as testing models, including VGG-16 (Simonyan & Zisserman, 2014), ResNet-50 (He et al., 2016), DenseNet-121 (Huang et al., 2017), and BN-Inception (Szegedy et al., 2016). Note that in the SM setting, the source model ResNet-50 is identical to the ResNet-50 model used in the testing models, which is known as a white-box attack.

Due to space constraints, the experimental setups, details of attack algorithms, techniques for managing perturbations, and experimental results are shown in Appendix B. The experimental setup for the MNIST and CIFAR-10 datasets is listed in Appendix C.

## 5 EXPERIMENTAL RESULTS

### 5.1 TYPES OF HUMAN-IDENTIFIABLE FEATURES

To gain a better understanding of the paper, we introduce two types of human-identifiable features based on our experimental results: the masking effect and the generation effect.

For the masking effect, the perturbations mimic specific features in the input image but are inverted to act as the negative counterparts of those features. When these perturbations are combined with the original image, they lower the contrast in pixel values of key classification features. This reduces the value of the inner product between the image and the gradient of its labeled class score, thus increasing the likelihood of misclassification. The "masking effect" usually occurs in untargeted attacks.

The generation effect is characterized by the injection of additional features into the original image. These features closely imitate the attributes of a different class, causing human observers and neural networks to misclassify the image into an incorrect class. This effect is primarily observed in targeted attacks.

---

[1]The selected classes are: great white shark, cock, tree frog, green mamba, giant panda, ambulance, barn, baseball, broom, bullet train, cab, cannon, teapot, teddy, trolleybus, wallet, lemon, pizza, cup, and daisy.

## 5.2 UNTARGETED ATTACK

Figure 2 shows three images from the ImageNet dataset with their corresponding adversarial perturbations generated in the SM and MM+G settings for BIM, CW, and DeepFool attack algorithms under the untargeted attack mode.

In the SM setting, the perturbation contains a large amount of noise and incomplete information that obscures, as an example, the features of a shark in the perturbation, shown in the middle of Figure 2. In the MM+G setting, in comparison, the perturbation reveals an identifiable feature, the shark contour, which resembles that of the original image. This observation supports our assumptions that human-identifiable features can be hidden within perturbations. Due to background noise as well as its incomplete feature, the shark may not appear noticeable to human eyes. Averaging perturbations from different models effectively minimizes noise and puts together incomplete parts of the features, leading to the emergence of pronounced human-identifiable features.

The phenomenon that perturbations contain features that resemble those of the original image with a flip sign, demonstrated in Figure 2, is the masking effect, as discussed previously. We emphasize that the masking effect is consistently observed in perturbations generated under the MM+G setting throughout our experiments. Please refer to Appendix D for additional examples of the masking effect.

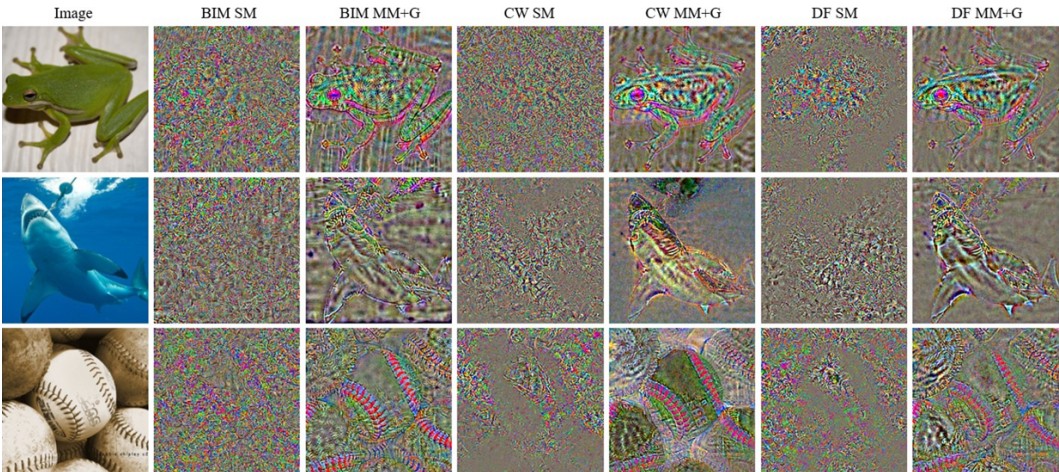

Figure 2: Adversarial perturbations generated by untargeted attacks in SM and MM+G settings. In the SM setting, perturbations appear as noise when viewed by humans. On the contrary, in the MM+G setting, however, perturbations reveal clear, human-identifiable features that resemble those in the original images, which is the masking effect mentioned in the text.

### 5.2.1 EVALUATING RECOGNIZABILITY OF PERTURBATIONS

In this experiment, we assess the recognizability of human-identifiable features in adversarial perturbations by conducting evaluation tests on all perturbations generated in the previous experiment. For enhanced visualization, the perturbations are linearly scaled according to the method outlined in Appendix B.2. Since the masking effect produces features that closely mirror those in the original images, we should be able to infer the original image's label from the generated perturbations if these features are distinct enough. Therefore, in the human evaluation test, we assign the label of each perturbation's corresponding image as the correct answer.

For the human evaluation test, we randomly divided the 200 perturbations into four equal subsets. In each subset, twelve different participants were tasked with assigning the most appropriate label to each perturbation, selecting from a predefined list of 20 classes in our experimental dataset. After discarding the highest and lowest classification accuracy in each subset, we calculated an average accuracy of 80.7% for the BIM attack algorithm under the MM+G setting. This high level of accuracy suggests that the features introduced by the masking effect are indeed highly recognizable for humans. For reference, random guessing yields a 5% accuracy rate.

Due to resource constraints, a comprehensive human evaluation across all settings (SM, MM+G) and three attack algorithms was not feasible. Instead, we employed the VGG-16 model, which has not been included in our source models, for machine evaluation. The settings for the evaluation mirrored those for human testing, except we multiplied the perturbations by 0.5 to mitigate the effect posed by domain shifting, thereby improving the model classification accuracy. Additionally, we designated the model's output class as the one with the highest output value among the 20 pre-defined classes, consistent with the method used in the human evaluation test.

In the SM setting, VGG-16's accuracy is 5.5%, 4.5%, and 5.0% for BIM, CW, and DeepFool attacks, respectively, which is roughly equivalent to random guessing. On the contrary, VGG-16 achieved accuracies of 56.0%, 38.0%, and 38.0% for BIM, CW, and DeepFool attack algorithms, respectively. It is worth noting that the model's classification performance was much less accurate than that of human participants, largely due to the domain shift introduced by classifying perturbations instead of images. The fact that both human and machine classifiers significantly outperform random guessing suggests that the perturbations contain features that are crucial for classifying the original images.

### 5.2.2 Evaluating Attack Strength

To verify that adversarial perturbations obtained from the MM+G setting can still attack neural networks, we evaluated their attack strength. We used 200 adversarial perturbations generated in the experiment under both SM and MM+G settings. Then, we processed them by multiplying a quantity $\varepsilon$ with their signs, as shown below:

$$V' = \varepsilon \cdot sign(V) \tag{2}$$

Here $V$ and $V'$ are the original and processed perturbations, respectively. $\epsilon = 0.02$ for all attack algorithms. This process is similar to the fast gradient sign method (Goodfellow et al., 2015). As a result, every perturbation has the same $L_{inf}$ and $L_2$ norm. Next, we incorporated the processed perturbations into the input images and sent them to testing models to evaluate their classification accuracy.

On average, the four testing models correctly classify 81.8% of the input images. However, in the SM setting, the incorporation of perturbations generated by the BIM attack lowers classification accuracy to 63.3%. In the MM+G setting, perturbations further reduce classification accuracy to 13.2% for the BIM attack. Similar outcomes have also been observed from the CW and DeepFool attacks. This confirms the strong attack ability of perturbations generated within the MM+G setting, suggesting that the essence of perturbations remains unaltered after processing by Eqn. 1. For comprehensive information on attack accuracy, please consult Appendix E.

### 5.2.3 Experiment on Contour Extraction

In this section, our objective is to ascertain whether human-identifiable features serve as a key factor in fooling neural networks. We first designate the area within the contour of labeled objects in the input image as the "contour," and the area outside of this contour as the "background". Since precisely defining human-identifiable features is a complex task, we opt for a more inclusive definition. Thus we redefine the contour part of perturbations as human-identifiable features. This is based on the observation that humans classify images primarily based on internal contour information rather than background information. We then use pixel-level annotation from the ImageNet-S dataset Gao et al. (2022) to extract the contour part of the perturbations. A visual inspection of the contour extracted from perturbations identifies a good alignment with the human-identifiable features. For examples showcasing the results, please refer to Appendix F.

After separating contours and backgrounds from 200 adversarial perturbations, we process them with Eqn. 2, keeping $\epsilon = 0.02$. These perturbations are applied to the original images and tested across four models. The results show that contour-only perturbations, generated by BIM, CW, and DeepFool algorithms, dramatically decrease the average model accuracy from 81.8% to between 32.2% and 37.0%. In comparison, perturbations affecting only the background result in mild drops in accuracy, between 60.6% and 69.0%. It is important to note that the areal ratio of the contour to the background in the images stands at 0.83. This suggests that even if the area covered by the perturbed contour and its associated $L_2$ norm are comparatively smaller than those of the background, the contour's ability to facilitate effective attacks is substantially greater.

In our subsequent analysis, we vary $\epsilon$ from 0.01 to 0.1 in Eqn. 2 at 0.01 intervals, altering the $L_{inf}$ norm of the contours and backgrounds to observe changes in testing models' classification accuracy. In Figure 3, the x-axis is the perturbation's $L_{inf}$ norm, while the y-axis shows the average accuracy over four testing models. Data points labeled only by the attack algorithm denote perturbations with contours being extracted, whereas those labeled by both the attack algorithm and 'BG' denote background extraction. The differences in model accuracies between contour removal and background removal in perturbations displayed in Figure 3 again demonstrate that human-identifiable features are critical to adversarial attacks.

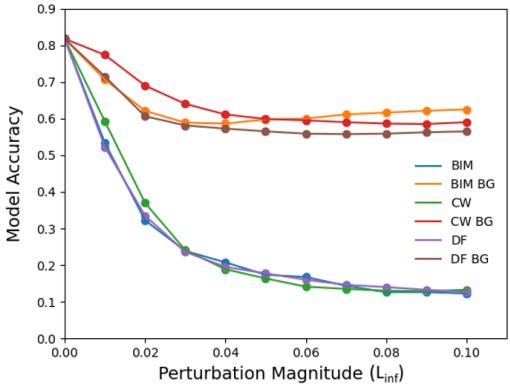

Figure 3: Effect of varying $\epsilon$ on perturbations' contour and background attack strength.

### 5.2.4 SEARCH-BASED ATTACKS

Previous experiments in this paper focused on gradient-based attacks. Here we examine whether search-based attacks like Square attack and One-pixel attack also generate perturbations containing human-identifiable features. The experimental setup remains the same as before, except we skip the process of adding Gaussian noise to input images since Square attack and One-pixel attack are inherently stochastic.

Random search-based algorithms are less efficient than backpropagation in terms of optimization capability, and they require more computational resources to produce pronounced human-identifiable features. Consequently, we limited our study to a subset of the 200 images used previously. parameter details for the attack algorithms are listed in Appendix G.

In Figure 4, we show the perturbations generated under the MM+G setting from the square attack and One-pixel attack. Figure 4 shows that search-based attack algorithms also reveal the masking effect, even in the absence of Gaussian noise as long as a sufficient number of perturbations are averaged. This strengthens our argument that adversarial perturbations contain human-identifiable features. More images are shown in Appendix D.

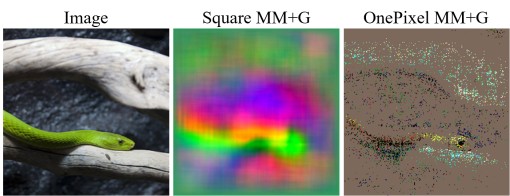

Figure 4: Perturbations, generated by Square attack and One-pixel attack, averaged across different models under the MM+G setting.

### 5.2.5 CONVERGENCE OF PERTURBATIONS

Our assumptions suggest that, under the untargeted attack and the MM+G setting, perturbations of the same image from different attack algorithms are likely to be similar. This is because the masking effect reduces contrasts on key features of an image and those key features should not be dependent on the attack algorithm. Furthermore, noise has near-zero similarity between different perturbations, which reduces overall similarity among perturbations. In contrast, in the MM+G setting, noise is removed through averaging. As a result, we expect increased similarity between perturbations of the same image across different attack algorithms. To quantify the similarity, we compute the cosine similarity between adversarial perturbations produced by different attack algorithms for the same image under the MM+G setting.

The calculation of cosine similarity is based on the contour part of the perturbations, which can be extracted using pixel-level labels from the ImageNet-S dataset, as described in Section 5.2.3. We then averaged these cosine similarity values across all 200 perturbations used in the experiment.

The averaged cosine similarities are notably high, ranging between 0.43 and 0.64, which agrees with our expectations and is shown in Figure 5. The similarity matrix for the MM and SM settings is shown in Appendix A.3.

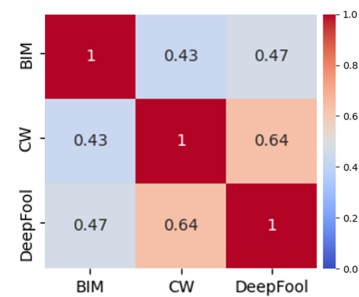

Figure 5: Cosine similarity for perturbations generated from different attack algorithms

## 6 TARGETED ATTACK

In the targeted attack mode, the generation effect becomes prominent. Although the experimental setup for targeted and untargeted attacks is the same, we use different image classes in the ImageNet dataset. This choice is dictated by the strong correlation between the class being attacked and the prominence of human-identifiable features in an image. For instance, between highly disparate classes—such as converting a car into a snake—different neural networks yield a variety of features due to the multiple feasible ways to make this transformation. Consequently, when averaging the perturbations generated by different models, the features tend to cancel out, making it challenging to obtain conclusive results. In contrast, when transforming closely related classes like hens into cocks, the required perturbations are more consistent across models. For example, adding a crest distinguishes a hen from a cock.

Figure 6 showcases adversarial examples generated by the CW attack under targeted attack mode.

For the two targeted attacks, the input images are labeled as Siamese cat and hen, and the target classes are designated as tiger and cock, respectively. Under the MM+G setting, we notice the change in the cat's fur to orange, more pronounced stripes, and a shift in eye color from blue to orange, all of which are features synonymous with a tiger, as shown in Figure 6(a). In Figure 6(b), the increase of the redness and the size of the crest on a hen are observed, making the hen resemble a cock. Additionally, a green patch appears below the chicken's head, and its feathers take on more brilliant hues, all are typical cock traits. Both examples reinforce our hypothesis that adversarial perturbations carry human-identifiable features. The generation effect is subtler than the masking effect. We contemplate that this subtlety is due to the lack of a standard transformation in the generation effect. This subtlety is also reflected in the challenge of transferring targeted attacks across diverse models, as noted in Liu et al. (2017) and Wang et al. (2023).

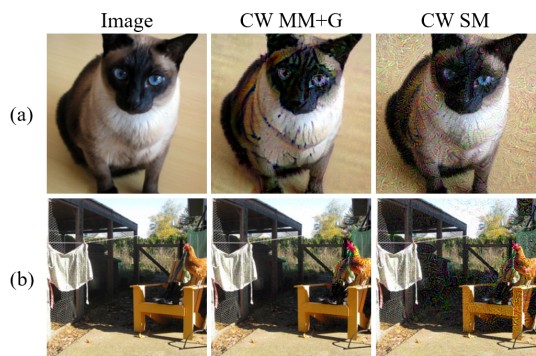

Figure 6: Examples of adversarial examples generated from targeted attack algorithm under MM+G and SM settings: (a) Transforming a Siamese cat to a tiger. (b) Transforming a hen to a cock. Under the MM+G setting, the tiger-like traits and the features of a cock become more pronounced, demonstrating the effect of generation.

## 7 DISCUSSION

We have proposed that adversarial perturbations contain human-identifiable features, which in turn contribute to the misclassification by neural networks. Based on this concept, three important phenomena of adversarial perturbations can be explained:

(1) Transferability: Researchers have found that a single perturbation can deceive multiple neural networks, extending beyond the model from which it was originally generated (Szegedy et al., 2014; Papernot et al., 2016). According to research on explainable AI, the features used by neural networks for classification may align with those used by humans (Smilkov et al., 2017; Selvaraju et al., 2017). In our study, we have found that adversarial perturbations modify features in the original image that are critical for humans to classify. Since neural networks may rely on human-identifiable features for classification and adversarial attacks alter some or all of these features, as perturbations are derived from different models, there is a likelihood that altered features overlap in different perturbations. As a result, perturbations are capable of being transferable across various models.

(2) Improving neural networks' interpretability via adversarial training: By utilizing adversarial examples in the training process, adversarial training enhances the interpretabilities of neural network gradients as well as perturbations. This, in turn, makes network decision-making mechanisms easier to understand. (Ross & Doshi-Velez, 2018; Tsipras et al., 2019; Santurkar et al., 2019).

In Section 3.2, we have pointed out that human-identifiable features are difficult to observe primarily due to excessive noise in the gradient and incomplete feature information. In the following, we contemplate how adversarial training effectively mitigates these two issues, allowing the reappearance of human-identifiable features.

As long as the network's gradient of the loss function remains roughly the same when incorporating perturbations to an input image (Goodfellow et al., 2015), adversarial training, in essence, minimizes the $L_2$ norm of the gradient (Simon-Gabriel et al., 2019). Since noise increases the $L_2$ norm of the gradient without yielding any performance benefit, it will be minimized during training. Furthermore, reducing the $L_2$ norm of the gradient results in a more evenly distributed range of gradient values, which facilitates the integration of incomplete human-identifiable features.

To illustrate the above argument, we consider a simplified example using a linear classifier. If two values of input data are equal, the classifier's output remains the same as long as the sum of the corresponding weights is constant. The classifier can thus assign these weights arbitrarily, provided their sum remains unchanged. When minimizing the $L_2$ norm, the classifier aims to distribute the weights of the two values as evenly as possible. This avoids undue focus on specific regions while neglecting others, thus allowing for more complete information in the model's weights. Given the piecewise linear properties of neural networks, similar results can be extended to neural networks.

Consequently, similar to what we have demonstrated earlier with the MM+G setting, adversarial training not only reduces noise but also allows for more complete information in the gradient. This results in gradients/perturbations that are better aligned with human perception.

(3) Non-trivial accuracy for classifiers trained on a manipulated dataset: Researchers found that classifiers trained on a perturbed dataset through targeted attack and relabeled as the target class, surprisingly, have demonstrated high accuracy on a clean testing set (Ilyas et al., 2019). The underlying causes of such a phenomenon are not yet fully understood.

We provide a partial explanation for the phenomenon by pointing out that adversarial perturbations contain human-identifiable features. Even when the neural network is trained on seemingly incorrect labels, the perturbations still contain accurate human-identifiable features aligned with correct labels. As a result, the network can still be trained to correctly classify input data based on those human-identifiable features. This leads to a non-trivial classification accuracy on a clean dataset.

## 8 CONCLUSION

In this study, we make several noteworthy discoveries concerning adversarial perturbations. First, when we average perturbations over different neural networks for a single image, we find two types of features that are easily recognizable by humans. Second, the contour part of these perturbations is significantly more effective at attacking models than that of the background part. Third, when averaged across different neural networks, perturbations created by different attack algorithms show notable similarities. These findings support the idea that human-identifiable features are inherently embedded in a large class of adversarial perturbations. This insight enables us to explain three related phenomena. Our study shows that human-identifiable features play an important role in fooling neural networks, which has been overlooked in the literature.

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

## A  EXPERIMENTAL RESULT WITHOUT THE INCORPORATION OF GAUSSIAN NOISE

To determine whether the appearance of human-identifiable features is inherent to adversarial perturbations, rather than an artifact of Gaussian noise introduced in the MM+G setting, we carried out a noise-free experiment. In this test, we averaged perturbations from multiple models (referred to as the MM setting) under untargeted attack algorithms. We then repeated several experiments initially conducted in the MM+G setting to see if comparable results were obtained. Due to a limited number of source models, this investigation was limited to the ImageNet dataset.

### A.1  EMERGENCE OF HUMAN-IDENTIFIABLE FEATURES

As illustrated in Figure 7, our visual analysis reveals that generated perturbations still retain human-identifiable features similar to those in the original images, which is the masking effect. Importantly, this phenomenon can be observed across the experiment, not just in the examples shown in Figure 7. However, the human-identifiable features are less pronounced in comparison to those in the MM+G setting. The number of averaged perturbations in this setting is much smaller - about a tenth compared to MM+G - thus limiting the noise reduction in the perturbations. It therefore is expected that the human-identifiable features are less obvious compared to those from MM+G.

Notably, in the case of search-based algorithms, which inherently possess randomness, additive Gaussian noise when generating perturbations is not essential, as detailed in Appendix G. Nevertheless, we still observe a masking effect in the perturbations. Our observations indicate that this masking effect persists in both gradient-based and search-based attack algorithms, even without the addition of Gaussian noise. Therefore, we can infer that the presence of human-identifiable features is inherent in the perturbations themselves, rather than being a result of added Gaussian noise.

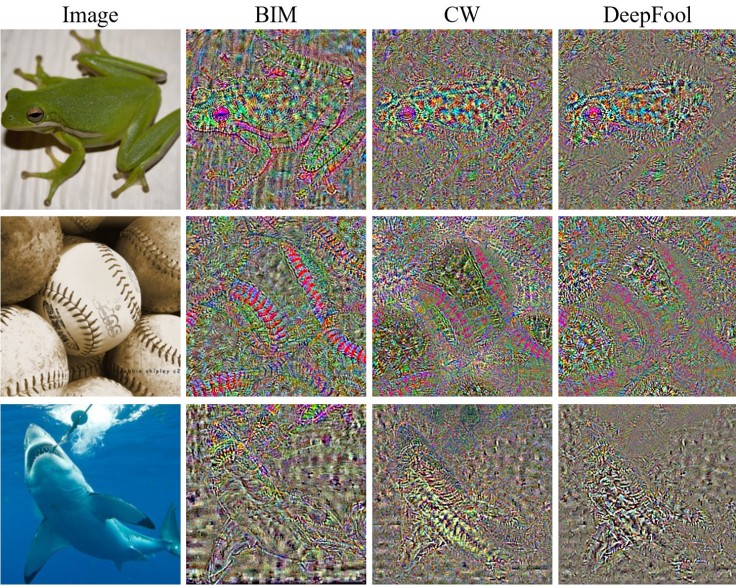

Figure 7: Adversarial perturbations generated via gradient-based algorithms in the MM setting for untargeted attacks. These perturbations maintain features that are similar to those in the original images, indicating an inherent masking effect not caused by Gaussian noise. While these features are less distinct than those in the MM+G setting, it is important to note that the number of perturbations used for averaging, in this case, is only one-tenth of that employed in the MM+G setting.

### A.2  CONTOUR EXTRACTION EXPERIMENT UNDER MM SETTING

In this experiment, we replicate the contour extraction experiment described in Section 5.2.3. However, this time the perturbations are generated using the MM setting.

The experiment results are as follows:

(1) The perturbations generated from contours are a key factor causing the model to misclassify. On average, across four testing models and 200 images, under the condition of $\epsilon = 0.02$, these perturbations lower the model's accuracy rate from 81.8% to between 32.9-39.2%. In contrast, when images include the background but exclude the contour component of perturbations, the accuracy stays within the range of 65.0% to 69.8%.

(2) Figure 8 plots the average model accuracy vs. $\epsilon$. Here, we vary the $L_{inf}$ norm of the perturbation contours and backgrounds by adjusting $\epsilon$ from 0.01 to 0.1 in increments of 0.01, as per Eqn. 2. Note that the experimental setup mirrors that in Section 5.2.3. We can see that as $\epsilon$ increases to 0.1, the attack strength of the contours drops down to 11.3%, whereas the strength of the background attack quickly saturates at 61.4%.

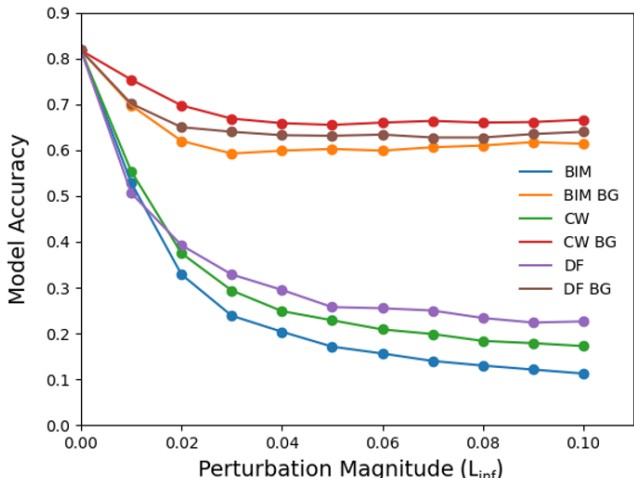

Figure 8: Effect of $L_{inf}$ norm variation on contour and background attack strength in the MM setting. The x-axis represents $\epsilon$, and the y-axis shows the average accuracy over four models and 200 test images. Data points labeled only by the attack algorithm denote perturbations with contours being extracted, whereas data points labeled by both the attack algorithm and 'BG' denote background extraction. As the $L_{inf}$ norm increases, the influence of background perturbations stabilizes (61.4-66.9%), whereas contour perturbations continue to substantially degrade model accuracy, dropping to 11.3% at $\epsilon = 0.1$.

The results from the MM setting align with those from the MM+G setting, i.e., showing notably stronger attack strength for contour perturbations compared to that of the background perturbations. These findings reaffirm that the observed strong attack strength from human-identifiable features is not due to the inclusion of noise in the MM+G setting.

## A.3 Convergence of Cosine Similarities

Finally, we repeat the experiment of calculating cosine similarities under the MM setting. We have found that the average cosine similarities of perturbations generated by different attack algorithms, after contour extraction, range between 0.40 and 0.58. This is higher than the cosine similarities of perturbations generated in the SM setting, ranging between 0.13 and 0.31, as shown in Figure 9. This is consistent with the results obtained from perturbations in the MM+G setting, shown in Figure 5.

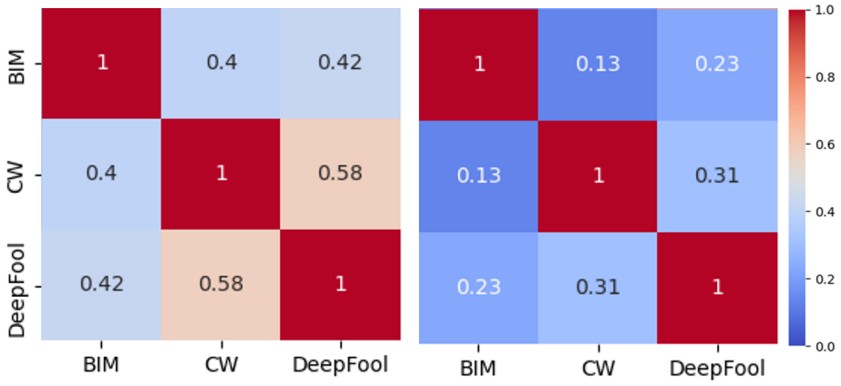

Figure 9: Measuring cosine similarity for perturbations in the MM (left panel) and SM (right panel) settings. For the MM setting, the average cosine similarities among various attack algorithms ranged from 0.40 to 0.58 after contours were extracted. Compared to the SM setting (0.13 to 0.31), these values are higher and agree with the results obtained from the MM+G setting.

Consistent findings between the MM+G and MM settings reinforce the idea that human-identifiable features are an inherent property of adversarial perturbations, rather than the consequence of added noise.

## B Detailed Information for Experimental Setup

### B.1 Parameters for Gradient-based Attacks

We analyze both BIM and CW attacks in both targeted and untargeted modes. The DeepFool attack is also investigated, but only in untargeted attack mode, as it does not support targeted attacks. BIM attack uses $\epsilon = 0.02$, $\alpha = 0.0008$, and 50 iterations. The CW attack is configured with $c$=5, $\kappa$=5, and 1,000 iterations. For the DeepFool attack, we set $\eta$ to be 0.02 and limited the number of iterations to 50. Please note that the above notations are consistent with those used in the original papers.

Due to the computational demands of applying DeepFool to ImageNet under the MM+G setting, we modify the attack by focusing only on the top 10 classes ranked by output logits, rather than evaluating all ImageNet classes, to identify the most appropriate target label. This adjustment resulted in a hundredfold reduction in computational time. All attack implementations were carried out using Torchattacks (Kim, 2020).

### B.2 Displaying Perturbations

After generating adversarial perturbations, we need to present them as images, which requires scaling. It is important to note that the mechanisms for targeted and untargeted attacks to fool neural networks are different, so the methods for displaying them also vary.

For untargeted attacks, our method is to invert the perturbation (multiplying by -1) to cancel out the negative sign brought on by the masking effect, as discussed in Section 5.1. We linearly adjust the perturbation's mean and standard deviation to match the average mean and variance of the ImageNet dataset. This step corrects for pixel value biases that cause color distortion after averaging the perturbations. It helps our observation of the masking effect and we expect the displayed perturbations to reveal human-identifiable features that resemble those from the original image.

Perturbations from targeted attacks often exhibit the generation effect, characterized by the injection of additional features into the original image, effectively changing its class. Therefore, it is more meaningful to look at adversarial examples instead of adversarial perturbations. To enhance visualization, we proportionally scale the perturbations up and set their maximum value to 0.5 before adding them to the image.

### B.3 CALIBRATING OUTPUT VALUES

Added Gaussian noise to the input image can alter the model's output, which in turn affects the generated adversarial perturbations. For instance, with the DeepFool attack, noise may cause the model to incorrectly classify the input image, thus preventing the algorithm from starting. Furthermore, certain algorithms target the class with the second-highest score. When noise is added, it can shift these score rankings, resulting in a change of the targeted class. To mitigate the effect of noise on perturbations, we have developed a method to calibrate the output values.

Our method first computes a calibration vector $calib.$, obtained by subtracting the output vector of the noise-augmented image $f(x + N(0, \sigma^2))$ from that of the original image $f(x)$. Then, while calculating the output values of models to generate perturbations $\delta(x)$, we add this calibration vector to counteract the effects of noise. The mathematical form is expressed in Eqn. 3. This approximation is justified by the local linearity property of neural networks (Goodfellow et al., 2015).

$$
\begin{aligned}
f'(x + N(0, \sigma^2) + \delta(x)) &= f(x + N(0, \sigma^2) + \delta(x)) + calib. \\
calib. &= f(x) - f(x + N(0, \sigma^2))
\end{aligned}
\tag{3}
$$

### B.4 THE EFFECT OF CLIPPING

In the experiment, we do not clip adversarial examples to a range between 0 and 1. The reason for this is that clipping enhances the recognizability of perturbations by making them resemble the original image, especially in untargeted attacks. We aim to ensure that any observed resemblance arises directly from the perturbations themselves, rather than from the method of displaying perturbations.

We demonstrate the idea as follows: If the original image has a pixel value of 1, then after clipping, the corresponding perturbation will have a value of zero or less. This ensures that the resulting adversarial example will not have a pixel value exceeding 1. When we later display the perturbation, we multiply it by -1 and scale up its value to counter the negative sign brought by the masking effect, as described in Appendix B.2. Thus if a pixel value is large, clipping ensures that the displayed perturbations will also have a large value for that pixel.

In Figure 10, the observed clipping effect is illustrated. The clipped perturbation displays brighter colors and its contours align more closely with that of the original image, compared to the unclipped version. Note that in this example, clipping is applied to each individual perturbation before averaging.

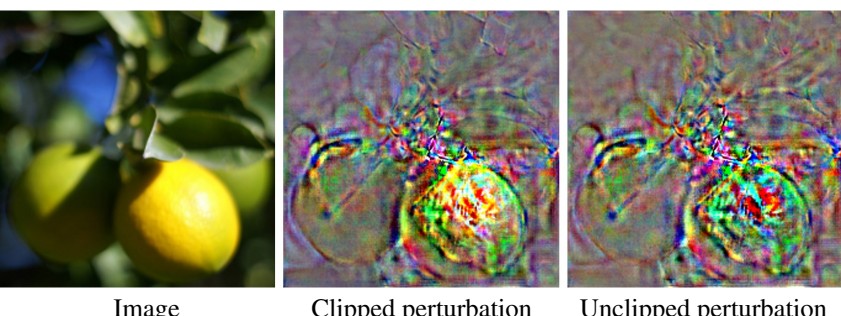

Image       Clipped perturbation       Unclipped perturbation

Figure 10: Clipping Effect for perturbations in MM+G Setting with BIM Attack. The panel shows a lemon and its clipped and unclipped perturbations. While the clipped version has colors and contours that are closer to the original image, this resemblance is artificially induced by the clipping operation.

# C  EXPERIMENTS ON MNIST AND CIFAR-10 DATASETS

## C.1  EXPERIMENTAL SETUP

In this section, we outline the experimental setup for the MNIST and CIFAR-10 datasets, highlighting how they differ from our experiments on the ImageNet dataset. While the overall experimental framework remains consistent across all datasets, there are dataset-specific variations in the methods for data selection, the employed models, and the parameters used. This information will be detailed in the following.

### C.1.1  DATA SELECTION

For CIFAR-10 and MNIST datasets, we selected the first 10 images from each of the 10 classes in the testing sets. This resulted in 200 distinct images. For these chosen images, we generate adversarial perturbations under both SM and MM+G settings.

### C.1.2  MODEL ARCHITECTURE

For the MNIST experiment, we used 101 self-trained models, all based on the same VGG architecture. These models differ only in their initialization value. Of these, 100 serve as source models while the remaining one serves as a testing model. In contrast, for the CIFAR-10 experiment, we employed the entire suite of models available on the PyTorchCV repository (Sémery, 2018), totaling 70 distinct models. We selected four distinct architectures—DenseNet-40, DIA-ResNet-164, PyramidNet-110, and ResNet-56—as our testing models. The remaining 66 models function as source models. It is worth noting that under the SM setting, we specifically chose ResNet-56 in the testing models as the source model.

### C.1.3  EXPERIMENTAL PARAMETERS

For the MNIST experiment, the BIM attack is configured with parameters $\epsilon = 0.2$, $\alpha = 0.008$, and the number of iterations is set to 50. In contrast, the CIFAR-10 experiment has different parameter values for the BIM attack: $\epsilon = 0.03$, $\alpha = 0.0012$, and the number of iterations remains the same. For both MNIST and CIFAR-10 experiments, the CW attack parameters are consistently set with $c = 0$, $\kappa = 0$, and a total of 1000 iterations. For the DeepFool attack, $\eta$ is 0.02, and the number of iterations equals 50 in both experiments.

For all experiments in the MM+G setting, the incorporated noise is sampled from an isotropic Gaussian distribution with a mean of 0. For the MNIST experiment, the Gaussian standard deviation is 0.2, and each image is copied 20 times, each incorporated with a different Gaussian noise, for every model. For the CIFAR-10 experiment, the standard deviation varies with attack algorithms. For both BIM and CW attacks, the standard deviation of noise is set at 0.05, and each image is copied 100 times with different Gaussian noises. Due to the time-consuming nature of the DeepFool attack, we reduced the number of copies to 20. Additionally, our empirical observations lead us to increase the standard deviation of Gaussian noise to 0.1 for DeepFool attacks on the CIFAR-10 experiment to further enhance the visibility of the generated adversarial perturbations.

C.2 EXPERIMENTAL RESULTS

Figure 11 demonstrates adversarial perturbations generated on CIFAR-10 and MNIST datasets under MM+G and SM settings. It is evident that under the SM setting for the MNIST dataset, the perturbations already exhibit distinct features recognizable by humans. These features become even more pronounced with reduced background noise under the MM+G setting, except for the CW attack where the SM setting already produces low noise and clear human-identifiable features. Interestingly, the MM+G setting also generates new, digit-like shapes in the perturbations, such as a blurred "4" for the digit "1" in the BIM attack and a clear "7" in the CW attack. This is the generation effect. Further examples of the generation effect are evident during targeted attacks.

MNIST

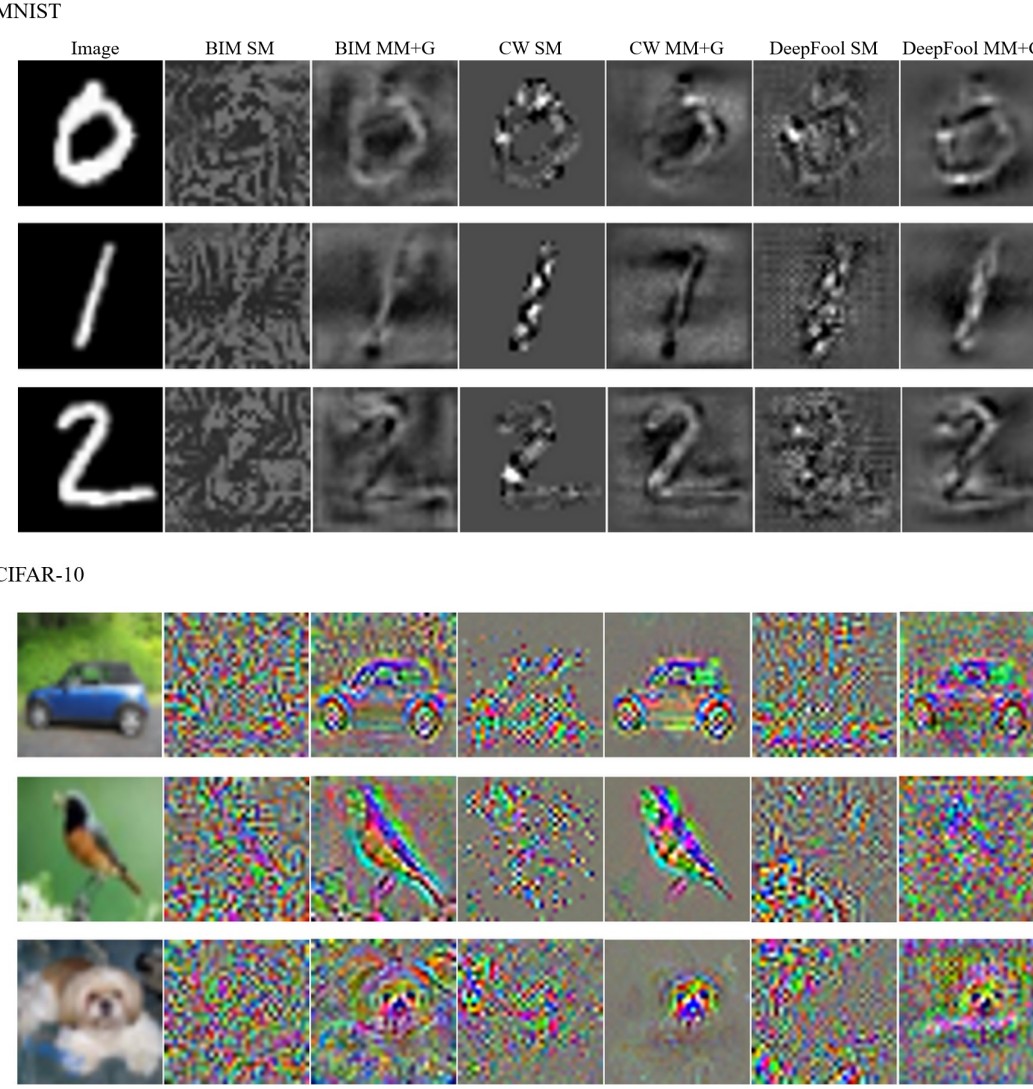

CIFAR-10

Figure 11: Adversarial perturbations generated under the MM+G setting for MNIST and CIFAR-10 datasets.

For the CIFAR-10 dataset under the MM+G setting, the perturbations for car and bird images exhibit identifiable features resembling the shapes of the input images. In contrast, the perturbations for the dog image primarily capture facial features. This suggests that human-identifiable features do not necessarily encompass an object's entire contour. While we present only three examples per dataset, it is important to note that the MM+G-derived perturbations consistently exhibit human-identifiable

features. This consistency is observed across 200 images used in the experiment and mirrors the masking effect seen in our ImageNet experiments.

### C.3    QUANTITATIVE ANALYSIS OF EXPERIMENTAL RESULTS

To assess the recognizability of the generated perturbations, we conducted a machine evaluation test. Specifically, we checked if the model's predictions for each scaled perturbation aligned with the label of its corresponding image. This method of evaluation was applied to all the generated perturbations. It is worth noting that our evaluation approach is analogous to the one used in the ImageNet experiment, as detailed in Section 5.2.1.

In the MNIST dataset, according to the testing model, 57%, 44%, and 68% of perturbations generated by BIM, CW, and DeepFool attacks under the MM+G setting are correctly classified, respectively. Comparatively, BIM, CW, and DeepFool attacks in the SM setting have accuracy of 18%, 46%, and 17%, respectively.

In the CIFAR-10 dataset, we averaged accuracy across four testing models. These models can classify MM+G-generated perturbations with 63%, 46%, and 21% accuracy using BIM, CW, and DeepFool attack methods, respectively. The classification accuracy for SM-generated perturbations is significantly lower, yielding accuracy nearly equal to random guessing at 9%, 10%, and 10%.

The strong performance of testing models over random guessing indicates that the perturbations contain features essential for accurately classifying the original images, which is described as the masking effect earlier. Additionally, the non-trivial accuracy observed in the MNIST dataset under the SM setting suggests that these perturbations already include observable identifiable features, in agreement with the observation in Figure 11.

# D MORE EXPERIMENTAL DATA

In Figure 12 and Figure 13, we supplement our study with five additional input images, each accompanied by adversarial perturbations from five different attack algorithms, under untargeted attack mode with both SM and MM+G settings. To offer a general overview of the dataset, we display the first image and its corresponding perturbations from each class, according to lexical file name order in the ImageNet dataset. Figure 12 and Figure 13 show perturbations from gradient-based attack algorithms and search-based attack algorithms, respectively. Consistent with our earlier findings, we observe a pronounced masking effect for perturbations generated under the MM+G setting.

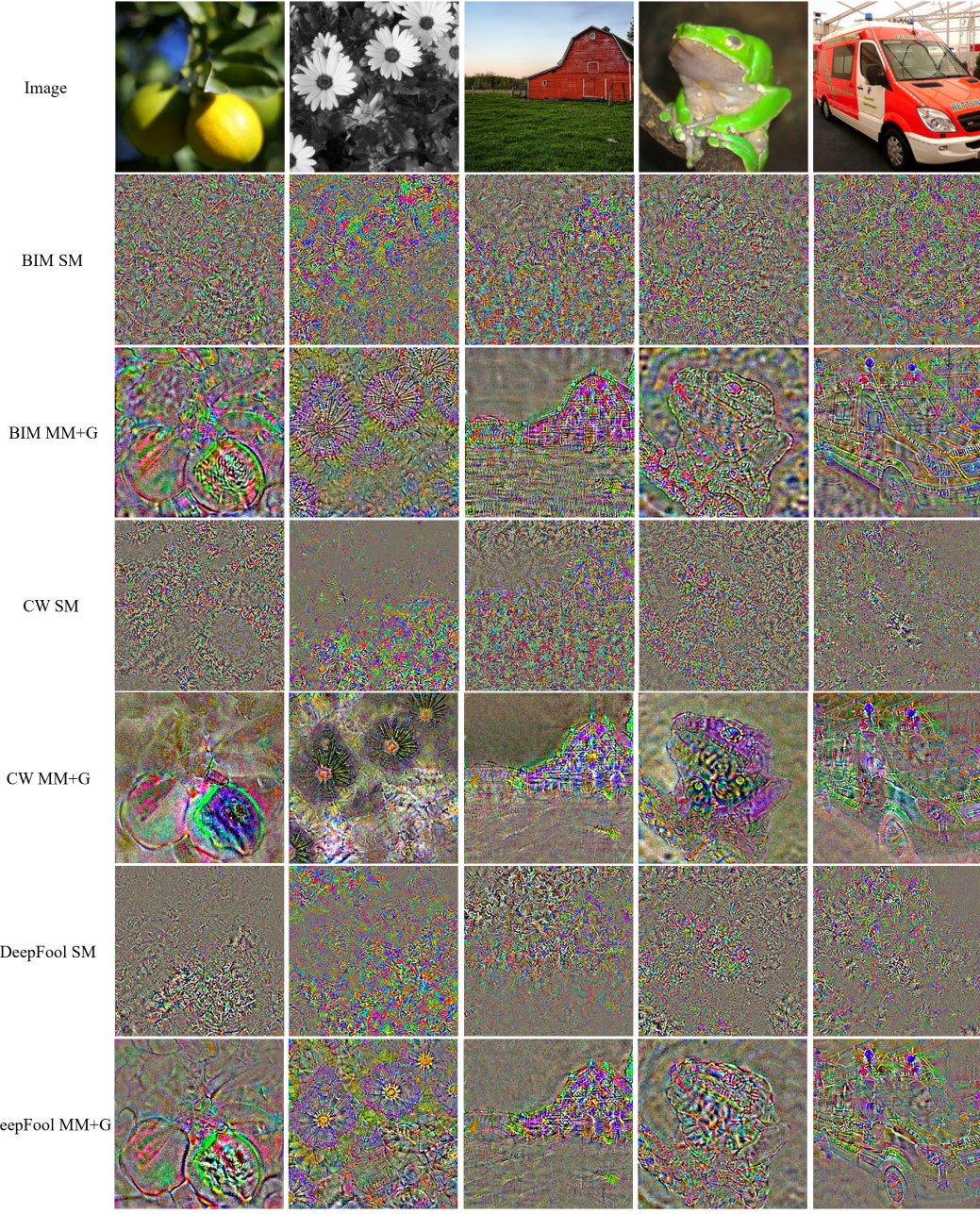

Figure 12: Additional examples of perturbations generated from gradient-based attacks.

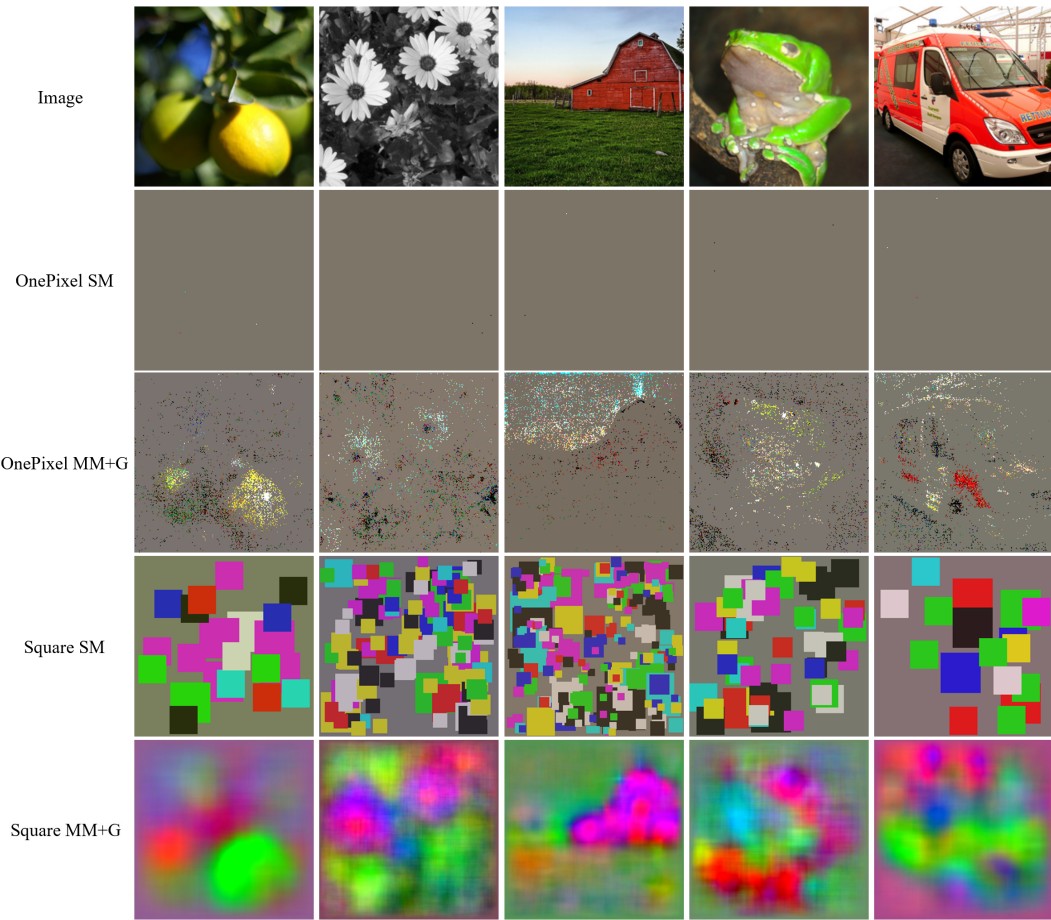

Figure 13: Additional examples of perturbations generated from search-based attacks.

## E  COMPREHENSIVE INFORMATION ON ATTACK ACCURACY

The attack strengths of the adversarial perturbations obtained from different attack algorithms for ImageNet are listed in Table 1. The Image column of Table 1 shows that, on average, the testing models correctly classify 81.8% of the input images. The effect of adding noise (Noise column), defined as random sampling where each pixel value has an equal probability of being +0.02 or -0.02, to images has a minimal effect on testing models' classification. The incorporation of perturbations in the SM setting lowers the classification accuracy to 63.3% for the BIM attack. In the MM+G setting, perturbations further reduce the classification accuracy to 13.2% for the BIM attack. This confirms the strong attack ability of perturbations generated in the MM+G setting.

Table 1: Attack strength of adversarial perturbations processed by MM+G.

| TESTING MODELS | REFERENCE | | SM | | | MM+G | | |
| --- | --- | --- | --- | --- | --- | --- | --- | --- |
| | IMAGE | NOISE | BIM | CW | DF | BIM | CW | DF |
| BN-INCEPTION | 81.5% | 83.0% | 64.0% | 77.0% | 68.0% | 16.5% | 22.0% | 15.0% |
| DENSENET121 | 83.5% | 83.5% | 58.5% | 77.5% | 66.5% | 10.5% | 16.5% | 13.0% |
| VGG-16 | 79.0% | 79.5% | 67.5% | 76.5% | 70.5% | 12.5% | 20.5% | 17.5% |
| RESNET50 | 83.0% | 82.0% | 0.0% | 7.0% | 4.5% | 13.0% | 18.5% | 14.0% |
| AVG. | 81.8% | 82.0% | 63.3% | 77.0% | 68.3% | 13.2% | 19.7% | 15.2% |

## F    CONTOUR EXTRACTION

Figure 14 compares the perturbations generated by the BIM attack in the MM+G setting and contour-extracted perturbations via pixel-level annotation of the image of a daisy. The extraction of contours leaves a homogeneous background, where the value is set to zero, in the perturbations.

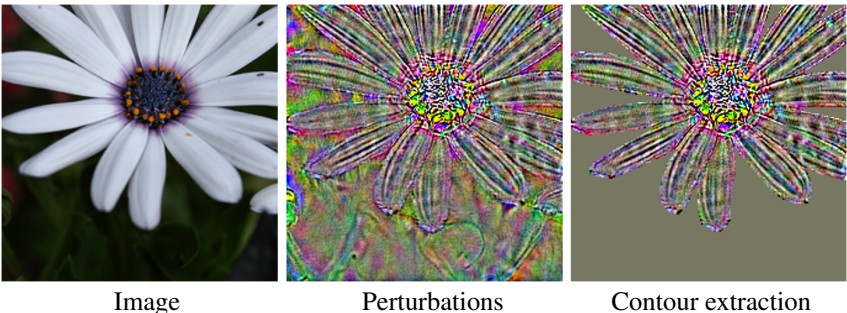

| Image | Perturbations | Contour extraction |

Figure 14: Visualizing contour extraction for a perturbation. From left to right, the input image of a daisy, perturbations generated by the BIM attack in the MM+G setting, and contour-extracted perturbations via pixel-level annotation.

## G    PARAMETERS FOR SEARCH-BASED ATTACKS

### G.1    SQUARE ATTACK

In the experiment, we chose the $L_{inf}$ norm-based Square attack algorithm. We modified the initial perturbation value, switching it from the original stripe pattern to zero. This adjustment was made to focus our study on the human-identifiable features that emerged during the perturbation optimization process, rather than the effects that originated from the stripe pattern. The attack parameters are specified as follows: The perturbation magnitude is assigned a value of $\epsilon = 0.05$, and the variable representing the size percentage, $p$, is set to 0.1. Under the MM+G setting, we conducted 100 attacks for each image utilized by a model.

To generate adversarial perturbations of a single image, we need to average approximately 27,000 perturbations. This means that generating a complete perturbation using an RTX 3090 graphics card would take about 5 hours. Due to limited computational resources, we generated adversarial perturbations for 40 images, comprising 2 images from each of the 20 classes used in the experiment.

### G.2    ONE-PIXEL ATTACK

For each attack, we executed 400 iterations using a population size of 200, targeting three pixels per perturbation. We performed 10 repeated attacks for each model with different initializations, averaging the generated perturbations for each input image. Due to the high computational demands, which require an average of 80 hours to generate a single perturbation in the MM+G setting on a NVIDIA Tesla V100 GPU, we restricted our MM+G experiments to 10 images, each from a different class[2].

## H    MORE EXPERIMENTAL RESULTS ON TARGETED ATTACK

In Figure 15, there are three input images labeled as Siamese cat, slug, and lemon with the targeted classes being tiger, snail, and orange, respectively.

Under the MM+G setting, Figure 15(a), demonstrates a change in the cat's eye color from blue to orange and the emergence of black stripes that are features synonymous with a tiger. For Figure

---

[2]Selected classes: ambulance, barn, baseball, daisy, green mamba, lemon, teapot, tree frog, great white shark, and cock.

15(b), the perturbations tend to transform the curled slugs into the shells of a snail. Meanwhile, in Figure 15(c), perturbations change the lemons' color from yellow to a more orange hue, making the adversarial examples resemble oranges.

The above finding holds for both CW and BIM attacks. However, under the SM setting, it is challenging to observe the generation effect for targeted attacks. The observations are consistent with those from Section 6.

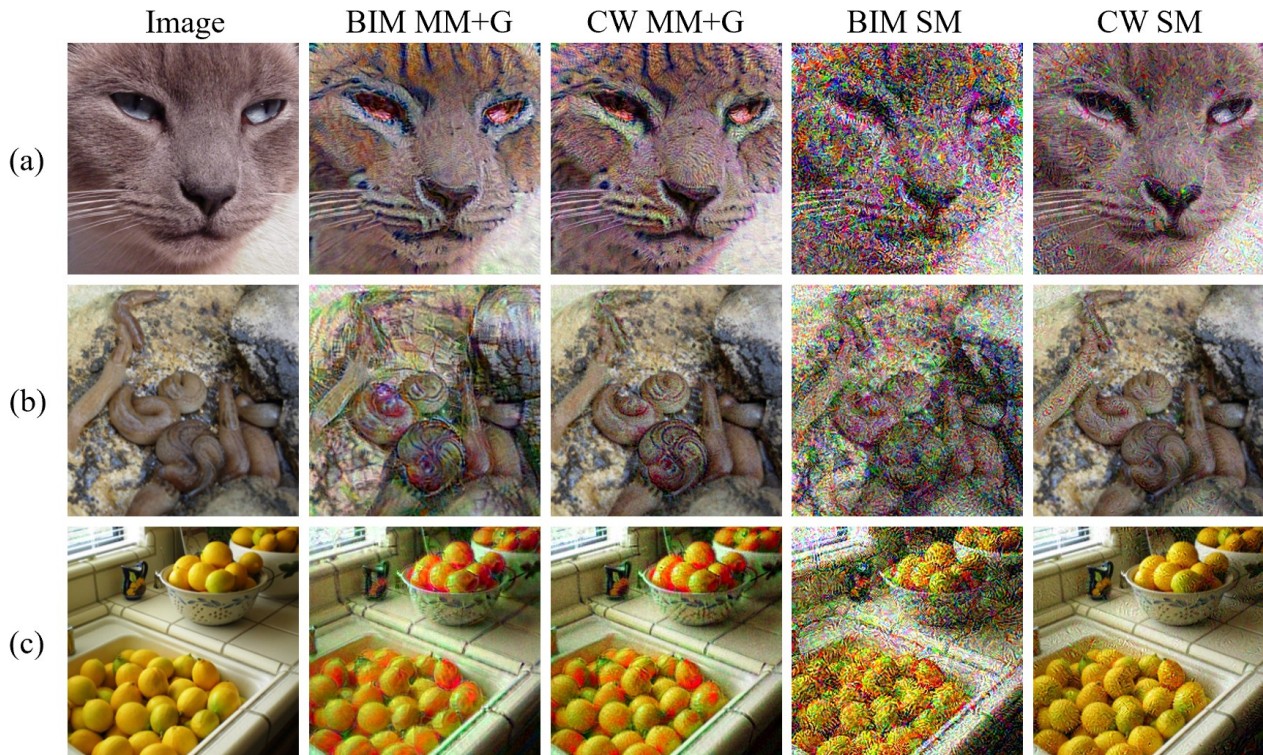

Figure 15: More examples of adversarial examples generated from targeted attack algorithm under MM+G and SM settings: (a) Transforming a Siamese cat into a tiger. (b) Transforming slugs into snails. (c) Transforming lemmons into oranges.

## I  MODEL COUNT AND THE MSE CONVERGENCE FOR PERTURBATIONS

This experiment investigates how the quantity of models employed for averaging perturbations influences the calculated mean square error (MSE) derived from 270 models in the MM setting. Our goal is to determine the number of models needed for the convergence of the MSE score, which is highly related to the emergence of human-identifiable features, and to understand how each model individually contributes to this MSE score.

Prior to calculating the MSE score, the averaged perturbations will be normalized using standard deviations and means from ImageNet datasets. This normalization ensures that the resulting MSE is on a comparable scale to the MSE scores calculated from images sampled from the ImageNet dataset, which have input values ranging between 0 and 1.

Figure 16 illustrates the outcomes of BIM, CW, and DeepFool attacks, showing the average MSE values between perturbations generated using varying numbers of models and those using 270 models. The x-axis represents the quantity of models employed in generating perturbations, and the y-axis shows the respective MSE values.

Figure 16 reveals a similar trend for perturbations from BIM, CW, and DeepFool attacks. To achieve MSE convergence within 0.05, an average of 25 models are needed for the three attack algorithms. For an MSE of 0.02, 90 models are required, while an MSE of 0.01 necessitates 157 models. The findings offer an estimate of the number of models required to attain MSE convergence in the MM setting, enhancing our understanding of how many models, on average, are necessary for the emergence of human-identifiable features.

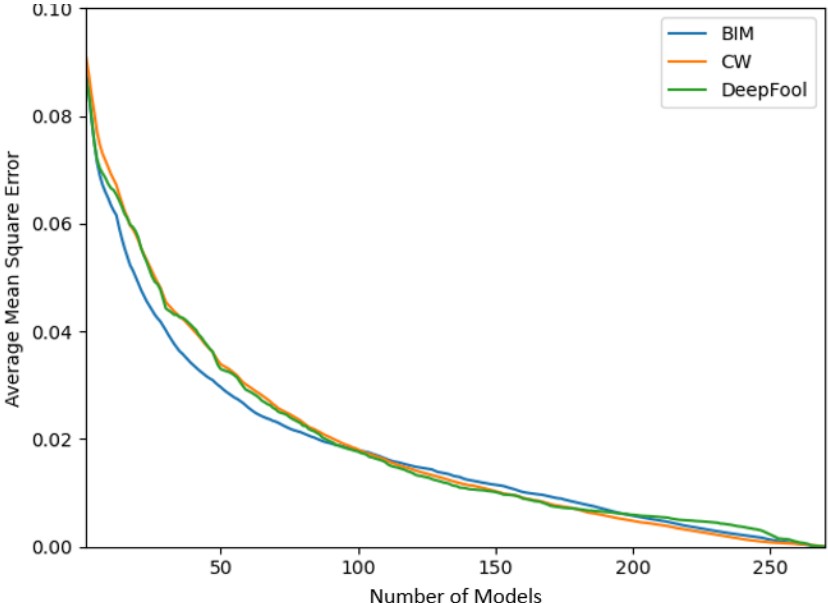

Figure 16: The averaged MSE scores between perturbations generated using varying numbers of models and those from the MM setting. The x-axis represents the quantity of models employed in averaging perturbations, and the y-axis shows the respective MSE scores.

