# OpenReview forum: "Investigating Human-Identifiable Features Hidden in Adversarial Perturbations"
_ICLR.cc/2024/Conference — Submitted to ICLR 2024_

### Official Review · Reviewer_MnMs · 2023-10-30

**Soundness:** 4 excellent
**Presentation:** 4 excellent
**Contribution:** 4 excellent
**Rating:** 10
**Confidence:** 5

**Summary:**

This work identifies the presence and effect of human-identifiable features in adversarial perturbations. The authors recognize that individual perturbations on a single input, while successful at fooling a model, do not produce distinct features that can be readily interpreted by humans. They posit that this is due to the presence of noise in the perturbations, and introduce a methodology to help overcome this by averaging many perturbations on the same image. The result produces perturbations that are significantly more human understandable, as demonstrated through a human evaluator experiment. With these new perturbations, they identify two different effects that these perturbations have on their input: masking, which covers prominent features of the true class of the image, and generation, which creates prominent features of the target class. Overall, this work provides insights into features created in adversarial examples, introduces methodology that can increase explainability in the presence of adversarial examples, and provides explanations from their findings for well known phenomena in adversarial training, transfer ability attacks, and interpretability.

**Strengths:**

Thank you for your submission! I thoroughly enjoyed reading this paper; the results were compelling, the methodology was sound, the contributions and findings are novel and useful, explanations were clear, and I was surprised at how recognizable the generated perturbations were.

Some specific highlighted results/conclusions/contribution:
- As mentioned in the paper, there is a significant need for work that provides explanations for reasons as to why attacks are as successful as they are and why models are as vulnerable to adversarial examples as they are. This work bridges these two approaches by (a) evaluating a variety of attacks and (b) creatively extracting portions of perturbations that are well aligned across models and thus represent features that transfer across models
- The perturbations generated with this method were significantly clearer/more recognizable to me as a reader. Additionally, I felt that the claim of generating human recognizable perturbations was well supported by also incorporating the results showing that (a) human evaluators were able to recognize perturbations without associated inputs from the MM+G method at a rate significantly higher than random guessing and (b) the perturbations generated in the MM+G setting yield far more successful adversarial examples than the standard SM case
- The discussion section connected multiple trends in transferability, adversarial training, and clean/robust accuracy tradeoffs to reasonable explanations based on insights from this work.

**Weaknesses:**

The breadth of experiments done was extensive, but I felt that in certain places, the depth of individual experiments could have been improved. Specifically:
- I would have preferred to see more samples per class evaluated (10 seems quite small to me)
- In the human evaluator test, I understand the limitation of testing all the attacks/settings but at the very least both settings under one attack should have been evaluated. At present, it is hard to give meaning to the 80.7% human evaluator accuracy under the BIM MM+G setting since there is not a BIM SM setting to compare it to. It would also be helpful to provide some justification for why BIM (over the other attacks) was chosen for this experiment.
- Similar to the previous point, including SM settings in the cosine similarity experiment would have been helpful to get a baseline sense of how similar perturbations usually are to each other and to see if the MM+G setting yields significantly different values.

Additionally, the paper is clear and concise as written, but there were some portions that could benefit from additional details, explanations, or citations, mainly in Section 4 (Experimental Method).

Specific (minor) suggestions for improvement:
- The notion of "incomplete components of the associated features" was lacking definition/explanation, adding some details around what this is supposed to represent would be helpful.
- The problem of "the number of available neural networks being limited" didn't feel clear/well motivated. There are many parameters that can be adjusted to produce different models (seeds, hyperparameters, optimizer, architecture, etc.). Further, it wasn't clear how the solution of applying noise to produce more inputs solved this problem.
- Some more citations to help support the contour extraction experiment would be helpful, particularly for claims that make statements about portions of the image that humans use for classification.

**Questions:**

- How were the subset of classes chosen?
- How were the 200 inputs chosen? Were there any constraints or conditions for these inputs? Were all samples chosen correctly classified by all models?
- While it does appear that adding noise to produce additional inputs works well, the inspiration/motivation for doing this wasn't exactly clear. Why add noise rather than performing some kind of data augmentation?
- Why was the standard deviation of noise added to the inputs different for the different attack algorithms?
- Why were 270 models chosen for generating perturbations? Were these experiments tried with fewer models (besides the single model case)?
- It is mentioned in the human evaluator test that the lowest and highest accuracy in each subset was discarded before calculating the average. What was the purpose of this? And can you clarify exactly what was discarded (e.g., was data for a single sample removed from all participants or was data from a single participant removed from all samples?)

---

> ### Author Response · Authors · 2023-11-23
> **Thank You for Your Comment**
>
> We wholeheartedly appreciate your encouragement and suggestions. Thank you very much.
>
>
> >How were the subset of classes chosen and 200 inputs chosen?
>
> We carefully selected 20 classes from the ImageNet dataset to ensure maximal diversity in our study, covering a wide array of subjects such as animals, plants, architecture, toys, transportation, and utilities. To avoid human bias, we adopted a systematic approach, selecting the first 10 images from each class rather than hand-picking them.
>
> >While it does appear that adding noise to produce additional inputs works well, the inspiration/motivation for doing this wasn't exactly clear. Why add noise rather than performing some kind of data augmentation?
>
> We believe that perturbations contain noise that may result in meaningless local variations [1]. As a consequence, introducing different Gaussian noises to the same input image may lead to different local variation patterns. Therefore, averaging those perturbations can effectively reduce the noise residing within them.
>
> We believe that the changes in local variations are independent of specific data augmentation techniques because their variation is meaningless. Consequently, we anticipate that different augmentation techniques will yield similar results. However, adding noise offers the advantage of easily controlling the strength of augmentation by adjusting its standard deviation. This is the reason why we choose to add noise instead of performing data augmentation.
>
> Reference:
>
> [1] Smoothgrad: Removing noise by adding noise. Daniel Smilkov, et al. Workshop on Visualization for Deep Learning. 2017.
>
> > Why was the standard deviation of noise added to the inputs different for the different attack algorithms?
> In our research, Gaussian noise serves as an additional tool to further reduce the noise in perturbations. We found that different attack algorithms may need different levels of Gaussian noise to result in optimal clarity for human-identifiable features.
>
> For BIM attacks, Gaussian noise with a standard deviation of 0.02 produces clear and pronounced human-identifiable features. CW and DeepFool attacks with a standard deviation of 0.02 also exhibit clear human-identifiable features. However, with a standard deviation of 0.05, the clarity of these features will be further enhanced. Consequently, we set the standard deviation for BIM attacks to 0.02 and for CW and DeepFool attacks to 0.05.
>
> We would like to emphasize that averaging perturbations from different models is sufficient for the emergence of human-identifiable features, as discussed in Appendix A. Adding Gaussian noise only serves as an additional tool to further reduce perturbation noise.
>
> >Why were 270 models chosen for generating perturbations? Were these experiments tried with fewer models (besides the single model case)?
>
> We have conducted additional experiments to investigate the impact of the number of models used for averaging perturbations on the mean squared error (MSE) compared to perturbations from the MM setting. This analysis may give us a better insight into the number of models required to observe the emergence of human-identifiable features.
>
> Prior to calculating the MSE score, we normalized the averaged perturbations using standard deviations and means derived from ImageNet datasets. This normalization ensures that the resulting MSE is on a comparable scale to the MSE scores calculated from images sampled from the ImageNet dataset, which have input values ranging between 0 and 1.
>
> Our findings reveal that to achieve MSE convergence within 0.05, an average of 25 models is required for the three attack algorithms. For an MSE of 0.02, 90 models are necessary, while an MSE of 0.01 needs 157 models. For detailed information, please refer to Appendix I.
>
> >It is mentioned in the human evaluator test that the lowest and highest accuracy in each subset was discarded before calculating the average. What was the purpose of this? And can you clarify exactly what was discarded (e.g., was data for a single sample removed from all participants, or was data from a single participant removed from all samples?)
>
> To eliminate outliers, we excluded data from two participants in each group: one with the highest accuracy and another with the lowest accuracy. This decision was made based on statistical considerations, as we observed that within each group, some individuals may perform significantly better or worse than the norm. By removing these outliers, we aim to provide an averaged human perspective on the evaluation of human assessment.

---

### Official Review · Reviewer_ZsSi · 2023-10-31

**Soundness:** 2 fair
**Presentation:** 1 poor
**Contribution:** 2 fair
**Rating:** 3
**Confidence:** 5

**Summary:**

This paper explores the human-identifiable features that are concealed within adversarial perturbations. To this end, this paper utilizes 270 models as surrogate models, introduces Gaussian noise to the input, and identifies the human-identifiable features. This paper shows that in targeted attacks, these features typically demonstrate a "generation effect" by producing features or objects of the target class. In contrast, in untargeted attacks, these features exhibit a "masking effect" by hiding the features or objects of the original class. This paper further claims the revealed phenomenon can interpret some properties of adversarial perturbations.

**Strengths:**

1. This paper revisits a critical concept in the context of adversarial robustness: the underlying mechanism of adversarial perturbations.
2. This paper conducted human tests to verify that the emergence of semantic features is not coincidental, which is of importance.
3. This paper validates the hypothesis across targeted and untargeted attacks and includes search-based attacks.

**Weaknesses:**

This paper challenges a well-acknowledged phenomenon in the context of adversarial robustness: the *perceptual aligned gradient* (PAG), which refers to the **human-identifiable features** that align with human perception in adversarial perturbations, only exists in robust models [1-3]. However, this paper claims that such features are also hidden in the perturbations of standardly trained (non-robust) models, which contradicts the current understanding of PAG. This concept of PAG has been well supported by various empirical and theoretical analyses in the follow-up works, along with its various applications. Therefore, in my opinion, to challenge the existing theories that contradict the claim made, this paper should provide sufficient theoretical and empirical evidence to support the proposed claims. Unfortunately, not only has the evidence in this paper already been discovered or directly deduced by previous work, but they also cannot explain the contradicted theories, which I specify below.

1. The experiment uses Gaussian noise to average the perturbations to reveal the human-identifiable features. However, this phenomenon has already been revealed in [4], which shows that randomized smoothing (adding Gaussian noises to the input and calculating the averaged gradient) on a single standardly trained model can lead to PAG and generate these features. Therefore, it's not a newly discovered phenomenon claimed in this paper that averaging gradient among perturbations with different noises can lead to human-identifiable features.
2. The experiment also averages different models to reveal the human-identifiable features. However, this phenomenon is expected based on existing work [5, 6], which shows that a little adversarial robustness of the models can lead to PAG. Specifically, as ensembling more non-robust models can still enhance adversarial robustness to a certain extent, though not as robust as adversarially trained models, it can be inferred that the ensembled model can lead to such PAG and identifiable features. Even if this paper shows that the robust accuracy of the ensembled model against adversarial attacks is still low (in Figure 3), the enhanced robustness may still be sufficient to bring such PAG.
3. In addition, it has also been shown [7] that the distribution of non-robust features [17] varies across different model architectures. Therefore, intuitively, the gradient (perturbation) of a single model (or a single kind of model architecture) may be noisy, but by averaging the gradients from different models, it is possible to converge toward the robust features.

Based on these discussions, the discovery made in this paper is somewhat trivial, since the observed phenomenons have already been revealed in existing work or can be directly deducted from them. Furthermore, the evidence presented in this paper is insufficient to challenge the well-established theories of PAG, as this paper does not provide a clear explanation of the contradictions or confusions, which I specify below.

4. There exist several works [8-10] aim to explain the reason PAG only exists in robust models by characterizing the decision boundaries between different models, which is well supported by theoretical analysis. These works show the fundamental difference of decision boundaries between standard and adversarially trained models leads to the (non-)existence of PAG, which contradicts the claim made in this paper in Section 7(2) that human-identifiable features also exist in non-robust models. Unfortunately, this paper does not discuss this viewpoint and does not conduct a theoretical analysis to overturn these theories.
5. There also exist theories interpreting the existence of PAG in robust models by modeling adversarial training as energy-based models [11-12]. Additionally, the robust model also provides better guidance during the generation process of diffusion models [13-14], indicating the importance of robust models with PAG for better gradient and generation guidance. Since such a generation process requires multi-step sampling, which can be regarded as applying an **average (ensemble)** of gradients (perturbations) to the standardly trained model, this also contradicts the viewpoint in this paper and should be well-explained.
6. In Section 7(1), the explanation for the transferability of adversarial examples contradicts existing works. This paper attributes the transferability to the human-identifiable (robust) features, but existing works [15-16] show that robust features may not be always helpful for adversarial examples transferring between models and non-robust features still play a crucial role in transferring adversarial examples. Therefore, the claims made in this paper fail to explain the transferability of adversarial examples across models.
7. The explanation of non-trivial accuracy for classifiers trained on a manipulated dataset [17] made in Section 7(3) is flawed. It is clear that in the manipulated dataset, which includes perturbations claimed as human-identifiable features in this paper, the features from the original class are still dominant over the perturbations. According to the interpretation within this paper, the model should still learn the features from the original class and cannot achieve clean accuracy in this noisy training setting. This contradicts the explanation proposed in this paper.
8. In Appendix A, Figure 7, it appears that the masking effect of the perturbation without Gaussian noise significantly reduces the identifiability of human-identifiable features, compared to the results in the main paper (with Gaussian noise). Therefore, it can be inferred that ensembling Gaussian noise plays a more crucial role in generating the human-identifiable features than ensembling different models, which undermines the soundness of the claim that the presence of human-identifiable features is inherent in the perturbations themselves, rather than being a result of added Gaussian noise.
9. There is a lack of ablation studies on the number of models to further support their claims. It is suggested to add experiments to analyze how many models or noises are required to emerge such human-identifiable features, which can provide a more intuitive view of how noisy the gradients are in the adversarial perturbations.
10. For transfer attacks, this paper only compares BIM, CW, and DF, which are not specifically designed for transfer attacks. It is suggested to add a comparison with existing state-of-the-art transfer attackers, e.g., MI-FGSM [18], DI-FGSM [19], and ensemble attacker CWA [20], to substantial the claims regarding transfer attacks. Since this paper claims that the success of transfer attacks is based on hidden human-identifiable features, it can be inferred that transfer attacks can emerge with more human-identifiable features, which should be supported by experiments on evaluating these attacks designed for transferring.
11. There is no statement on open sourcing and reproducibility. Since finding such 270 surrogate models is challenging to reproduce, I strongly suggest releasing the code.

[1] Robustness May Be at Odds with Accuracy. ICLR 2019

[2] Image Synthesis with a Single (Robust) Classifier. NeurIPS 2019

[3] Adversarial Robustness as a Prior for Learned Representations. arxiv 1906.00945

[4] Are Perceptually-Aligned Gradients a General Property of Robust Classifiers?. NeurIPS 2019 Workshop

[5] On the Benefits of Models with Perceptually-Aligned Gradients. ICLR 2020 Workshop

[6] A Little Robustness Goes a Long Way: Leveraging Robust Features for Targeted Transfer Attacks. NeurIPS 2021

[7] Skip Connections Matter: On the Transferability of Adversarial Examples Generated with ResNets. ICLR 2021

[8] Bridging Adversarial Robustness and Gradient Interpretability. ICLR 2019 Workshop

[9] On the Connection Between Adversarial Robustness and Saliency Map Interpretability. ICML 2019

[10] Robust Models Are More Interpretable Because Attributions Look Normal. ICML 2022

[11] Towards Understanding the Generative Capability of Adversarially Robust Classifiers. ICCV 2021

[12] A Unified Contrastive Energy-based Model for Understanding the Generative Ability of Adversarial Training. ICLR 2022

[13] Enhancing Diffusion-Based Image Synthesis with Robust Classifier Guidance. TMLR

[14] BIGRoC: Boosting Image Generation via a Robust Classifier. TMLR

[15] Closer Look at the Transferability of Adversarial Examples: How They Fool Different Models Differently. WACV 2023

[16] Why Does Little Robustness Help? Understanding and Improving Adversarial Transferability from Surrogate Training. S&P 2024

[17] Adversarial Examples are not Bugs, they are Features. NeurIPS 2019

[18]  Boosting adversarial attacks with momentum. CVPR 2018.

[19] Improving transferability of adversarial examples with input diversity. CVPR 2019.

[20] Rethinking Model Ensemble in Transfer-based Adversarial Attacks. arXiv:2303.09105

**Questions:**

Please see the weaknesses above.

---

> ### Author Response · Authors · 2023-11-22
> **We Respectfully Disagree With the Reviewer's Comment (Part 1)**
>
> We respectfully disagree with the reviewer's comment, yet we appreciate the reviewer's time. The reviewer clearly has an opposite view from ours on the origin of adversarial perturbations. This contrast in views, in our opinion, has led to a very critical tone in the reviewer’s comments. In the following, we respond to each of the reviewer's points, providing clarifications and counterarguments where necessary.
>
> Please note that references starting with R are referred to as the reviewer's references.
>
> >1. The experiment uses Gaussian noise to average the perturbations to reveal the human-identifiable features. However, this phenomenon has already been revealed in [R4], which shows that randomized smoothing (adding Gaussian noises to the input and calculating the averaged gradient) on a single standardly trained model can lead to PAG and generate these features. Therefore, it's not a newly discovered phenomenon claimed in this paper that averaging gradient among perturbations with different noises can lead to human-identifiable features.
>
> The reviewer confused our findings with those in [R4]. Our research reveals that human-identifiable (robust) features naturally occur in adversarial perturbations of 'standard-trained' neural networks, even without adding Gaussian noise. This stands in contrast to [R4], which associates perceptually aligned gradients with the robustness of a classifier, leading to a markedly different conclusion.
>
> A method for creating a robust classifier involves randomized smoothing, which adds Gaussian noises with a significant standard deviation (0.5 as mentioned in [R4]) to the original image and implicitly averages the model's predictions. The effectiveness of this technique lies in its certified radius, which assures the classifier's robustness within a certain range of perturbations and is proportional to the added Gaussian noises' standard deviation, as confirmed by earlier research work[1].
>
> In [R4], it is noted that as the standard deviation of incorporated Gaussian noise increases, human-perceptible features become observable and evident in perturbations. Hence, the author suggests that the increase of model's robustness may lead to the presence of perceptually aligned gradients.
>
> Our study, however, presents a different scenario. We demonstrate that human-identifiable (robust) features in adversarial perturbations are discernible when averaging perturbations from various models, even without Gaussian noise. This suggests that robust features inherently exist in standardly trained models, as averaging perturbations do not add new features to the original perturbation. Therefore, the distinction between our work and the study in [R4] is evident.

---

> > ### Author Response · Authors · 2023-11-22
> > **We Respectfully Disagree With the Reviewer's Comment (Part 2)**
> >
> > >2. The experiment also averages different models to reveal the human-identifiable features. However, this phenomenon is expected based on existing work [R5, R6], which shows that a little adversarial robustness of the models can lead to PAG. Specifically, as ensembling more non-robust models can still enhance adversarial robustness to a certain extent, though not as robust as adversarially trained models, it can be inferred that the ensembled model can lead to such PAG and identifiable features. Even if this paper shows that the robust accuracy of the ensembled model against adversarial attacks is still low (in Figure 3), the enhanced robustness may still be sufficient to bring such PAG.
> >
> > The reviewer states that according to prior research [R5 and R6], our findings are expected. However, our results are fundamentally different from those studies. In [R5], the author shows that adversarial training with low magnitude enhances the model's gradient perceptibility without significantly boosting its robustness, while [R6] discusses enhanced transferability of adversarial perturbations through a similar training method and is unrelated to our work.
> >
> > The reviewer, lacking concrete evidence, attributes the increased gradient perceptibility in [R5] to a marginal improvement in adversarial robustness, which is not stated in [R5]. The reviewer then states that our results are expected extension of [R5], based on the assumption that a slight increase in the model’s robustness will also increase perturbations’ perceptibility. There is no indication that such assumption can be realized. Verifying such assumption clearly does not fall within our work’s domain.  Furthermore, it is noteworthy that several defensive algorithms, such as defensive distillation [2], marginally enhance the model's robustness. We would be interested to know if the reviewer could offer thought on whether perturbations in these defensively augmented models exhibit a perception-aligned gradient.
> >
> > In our experiment, we average perturbations from different models and demonstrate that human-identifiable features naturally exist in perturbations from a standard-trained model. Averaging helps to reduce noise and put together incomplete information without incorporating new features into the perturbations. This should not be confused with an increase in models' robustness.
> >
> > >3. In addition, it has also been shown [R7] that the distribution of non-robust features [R17] varies across different model architectures. Therefore, intuitively, the gradient (perturbation) of a single model (or a single kind of model architecture) may be noisy, but by averaging the gradients from different models, it is possible to converge toward the robust features. Based on these discussions, the discovery made in this paper is somewhat trivial, since the observed phenomenon have already been revealed in existing work or can be directly deducted from them. Furthermore, the evidence presented in this paper is insufficient to challenge the well-established theories of PAG, as this paper does not provide a clear explanation of the contradictions or confusions, which I specify below.
> >
> > The paper in [R7] proposes an algorithm - the Skip Gradient Method (SGM) - that can increase the transferability of adversarial perturbations. In the paper, nothing is discussed about “the distribution of non-robust features”. We would appreciate it if the reviewer would show where the paper the claim is written.
> >
> > The reviewer initially mentioned that robust features exist exclusively in robust models, based on the concept of perceptual aligned gradients. Our research reveals, however, that robust features are also present in perturbations originating from standard-trained models, a finding established by averaging perturbations across various models. Our work is significant as it highlights the role of robust features for fooling models in perturbations from a standard-trained model. This insight has the potential to foster a deeper understanding of how perturbations can mislead models. Despite described clearly in the paper, the reviewer persistently extrapolates other studies and attributes our findings simply to increased model robustness, which deviates from the actual explanation and facts.

---

> ### Author Response · Authors · 2023-11-22
> **We Respectfully Disagree With the Reviewer's Comment (Part 3)**
>
> >5. There also exist theories interpreting the existence of PAG in robust models by modeling adversarial training as energy-based models [R11-R12]. Additionally, the robust model also provides better guidance during the generation process of diffusion models [R13-R14], indicating the importance of robust models with PAG for better gradient and generation guidance. Since such a generation process requires multi-step sampling, which can be regarded as applying an average (ensemble) of gradients (perturbations) to the standardly trained model, this also contradicts the viewpoint in this paper and should be well-explained.
>
> In our research, we show, without a shred of doubt, that human-identifiable features exist in perturbations of standard-trained models. We firmly believe that theory is always capable of explaining experimental findings.
>
> >6. In Section 7(1), the explanation for the transferability of adversarial examples contradicts existing works. This paper attributes the transferability to the human-identifiable (robust) features, but existing works [R15-R16] show that robust features may not be always helpful for adversarial examples transferring between models and non-robust features still play a crucial role in transferring adversarial examples. Therefore, the claims made in this paper fail to explain the transferability of adversarial examples across models.
>
> In our research, we isolated human-identifiable features from perturbations and discovered their significant attack strength, leading us to conclude that these features contribute to the misclassification of a model. We see no conflict between our findings and those in [R15, R16], as [R15, R16] do not deny the strong transferability of robust features to deceiving models.
> >7. The explanation of non-trivial accuracy for classifiers trained on a manipulated dataset [R17] made in Section 7(3) is flawed. It is clear that in the manipulated dataset, which includes perturbations claimed as human-identifiable features in this paper, the features from the original class are still dominant over the perturbations. According to the interpretation within this paper, the model should still learn the features from the original class and cannot achieve clean accuracy in this noisy training setting. This contradicts the explanation proposed in this paper.
>
> The term "dominant" should be used with caution. While the original images' features are larger in pixel value, the perturbations generated by the models may align more closely with what the models learn, potentially playing a more significant role during training than that of the original features. The discussion of which feature dominates in training is not the focus of this research. Since we have yet to demonstrate what types of features are more dominating, we put an emphasis in our paper that only partial explanation can be offered.

---

> ### Author Response · Authors · 2023-11-22
> **We Respectfully Disagree With the Reviewer's Comment (Part 4)**
>
> >8. In Appendix A, Figure 7, it appears that the masking effect of the perturbation without Gaussian noise significantly reduces the identifiability of human-identifiable features, compared to the results in the main paper (with Gaussian noise). Therefore, it can be inferred that ensembling Gaussian noise plays a more crucial role in generating the human identifiable features than ensembling different models, which undermines the soundness of the claim that the presence of human-identifiable features is inherent in the perturbations themselves, rather than being a result of added Gaussian noise.
>
> The statement made by the reviewer that the presence of human-identifiable features is due to the incorporation of Gaussian noise is incorrect. Two pieces of information illustrate the incorrectness:
>
> 1. In the search-based attack, no Gaussian noise is added, yet human-identifiable features still emerge, as shown in Section 5.2.4 and Appendix D.
> 2. In Appendix A, we have performed an experiment on perturbations generated without the incorporation of Gaussian noise (MM setting). It is clear that human-identifiable features is still observed from perturbations, as demonstrated in Appendix A1.
>
> Additionally, we disagree with the reviewer's assertion that averaging perturbations from various models (MM setting) significantly diminishes the perceptibility of human-identifiable features. The cosine similarity between perturbations generated separately from MM+G and MM setting for BIM attack reaches as high as 0.80 based on an average of 200 perturbations. This high similarity indicates that Gaussian noise is not critical to influencing the perceptibility of perturbations.
>
> > 9. There is a lack of ablation studies on the number of models to further support their claims. It is suggested to add experiments to analyze how many models or noises are required to emerge such human-identifiable features, which can provide a more intuitive view of how noisy the gradients are in the adversarial perturbations.
>
> We have added experiments analyzing the model count and the convergence for perturbations, please refer to Appendix I. Thank you for your suggestion.
>
> References:
>
> [1] Certified Adversarial Robustness via Randomized Smoothing. Jeremy Cohen, et al. ICML 2019.
>
> [2] Distillation as a Defense to Adversarial Perturbations against Deep Neural Networks, Nicolas Papernot, et al. 37th IEEE Symposium on Security and Privacy.

---

> ### Comment · Reviewer_ZsSi · 2023-11-22
> **Further response to the authors (Part 1/n)**
>
> Dear authors,
>
> Thank you for your response. I appreciate your time. However, I respectfully maintain my disagreement with your clarifications, which I will explain in detail below.
>
> ---
>
> 1. *The experiment uses Gaussian noise to average the perturbations to reveal the human-identifiable features. However, this phenomenon has already been revealed in [4], which shows that randomized smoothing (adding Gaussian noises to the input and calculating the averaged gradient) on a single standardly trained model can lead to PAG and generate these features.*
>
> I maintain my viewpoint on this concern. I believe that your clarifications have not yet distinguished your method from [4].
>
> > Our research reveals that human-identifiable (robust) features naturally occur in adversarial perturbations of 'standard-trained' neural networks, even without adding Gaussian noise. This stands in contrast to [R4], which associates perceptually aligned gradients with the robustness of a classifier, leading to a markedly different conclusion.
>
> While I acknowledge that [4] focuses on randomized smoothing in [21], the leveraged methods are exactly the same. Specifically, adding Gaussian noise to the sample and averaging the adversarial perturbations, both used in randomized smoothing [4] and in your proposed ``+G``, have the exact same underlying mechanism. Additionally, the model used in [4] is also a standardly trained model, which is still the same as your setting. Therefore, it is difficult to argue that your study presents a different scenario. The claim that adding Gaussian noise and then averaging over adversarial perturbations on standardly trained models can derive robust features has already been discovered.
>
> ---
>
> 2. *The experiment also averages different models to reveal the human-identifiable features. However, this phenomenon is expected based on existing work [5, 6], which shows that a little adversarial robustness of the models can lead to PAG. Specifically, as ensembling more non-robust models can still enhance adversarial robustness to a certain extent, though not as robust as adversarially trained models, it can be inferred that the ensembled model can lead to such PAG and identifiable features. Even if this paper shows that the robust accuracy of the ensembled model against adversarial attacks is still low (in Figure 3), the enhanced robustness may still be sufficient to bring such PAG.*
>
> > The reviewer then states that our results are expected extension of [5], based on the assumption that a slight increase in the model’s robustness will also increase perturbations’ perceptibility. There is no indication that such assumption can be realized. Verifying such assumption clearly does not fall within our work’s domain.
>
> Still, I maintain my viewpoint that ensembling multiple models into one can improve robustness, albeit not significantly, and thus improve perceptibility.
>
> First, it has been demonstrated that model ensemble can enhance adversarial robustness [22-23], thus leading to robust features. While the mentioned works focus on adversarially trained models, their assertion about the connection between ensembling and adversarial robustness can be easily extended to non-robust models.
>
> Furthermore, as stated in [5], the emergence of perceptual aligned gradients (for a single image, without being perturbed by noise) exhibits a strong correlation with model robustness. Therefore, since both methods of averaging over models and introducing perturbations contribute to improving robustness, recognized as randomized smoothing and model ensemble, these conclusions are undoubtedly anticipated.
>
> ---
>
> 3. *In addition, it has also been shown [7] that the distribution of non-robust features [17] varies across different model architectures. Therefore, intuitively, the gradient (perturbation) of a single model (or a single kind of model architecture) may be noisy, but by averaging the gradients from different models, it is possible to converge toward the robust features.*
>
> > The paper in [R7] proposes an algorithm - the Skip Gradient Method (SGM) - that can increase the transferability of adversarial perturbations. In the paper, nothing is discussed about “the distribution of non-robust features”. We would appreciate it if the reviewer would show where the paper the claim is written.
>
> The intuition here is that SGM [7] modifies the algorithm for generating adversarial examples based on the design of the architecture, which is specifically tailored for residual connection models. From this, it can be inferred that different model architectures exhibit different non-robust features. Intuitively, averaging various models leads to convergence towards robust features.
>
> ---
>
> It seems that you missed Weakness 4. I would like to further discuss with you after you fulfill
> > In the following, we respond to each of the reviewer's points, providing clarifications and counterarguments where necessary.
>
> Best Regards,

---

> ### Comment · Reviewer_ZsSi · 2023-11-22
> **References (to be updated)**
>
> [1] Robustness May Be at Odds with Accuracy. ICLR 2019
>
> [2] Image Synthesis with a Single (Robust) Classifier. NeurIPS 2019
>
> [3] Adversarial Robustness as a Prior for Learned Representations. arxiv 1906.00945
>
> [4] Are Perceptually-Aligned Gradients a General Property of Robust Classifiers?. NeurIPS 2019 Workshop
>
> [5] On the Benefits of Models with Perceptually-Aligned Gradients. ICLR 2020 Workshop
>
> [6] A Little Robustness Goes a Long Way: Leveraging Robust Features for Targeted Transfer Attacks. NeurIPS 2021
>
> [7] Skip Connections Matter: On the Transferability of Adversarial Examples Generated with ResNets. ICLR 2021
>
> [8] Bridging Adversarial Robustness and Gradient Interpretability. ICLR 2019 Workshop
>
> [9] On the Connection Between Adversarial Robustness and Saliency Map Interpretability. ICML 2019
>
> [10] Robust Models Are More Interpretable Because Attributions Look Normal. ICML 2022
>
> [11] Towards Understanding the Generative Capability of Adversarially Robust Classifiers. ICCV 2021
>
> [12] A Unified Contrastive Energy-based Model for Understanding the Generative Ability of Adversarial Training. ICLR 2022
>
> [13] Enhancing Diffusion-Based Image Synthesis with Robust Classifier Guidance. TMLR
>
> [14] BIGRoC: Boosting Image Generation via a Robust Classifier. TMLR
>
> [15] Closer Look at the Transferability of Adversarial Examples: How They Fool Different Models Differently. WACV 2023
>
> [16] Why Does Little Robustness Help? Understanding and Improving Adversarial Transferability from Surrogate Training. S&P 2024
>
> [17] Adversarial Examples are not Bugs, they are Features. NeurIPS 2019
>
> [18] Boosting adversarial attacks with momentum. CVPR 2018.
>
> [19] Improving transferability of adversarial examples with input diversity. CVPR 2019.
>
> [20] Rethinking Model Ensemble in Transfer-based Adversarial Attacks. arXiv:2303.09105
>
> [21] Certified Adversarial Robustness via Randomized Smoothing. ICML 2019.
>
> [22] Improving Adversarial Robustness via Promoting Ensemble Diversity. ICML 2019
>
> [23] Self-Ensemble Adversarial Training for Improved Robustness. ICLR 2022

---

> ### Author Response · Authors · 2023-11-23
> **Disagree With the Reviewer's Response (Part 1-2)**
>
> We respectfully disagree with the reviewer's comment, yet we appreciate the reviewer's time.
>
> >While I acknowledge that [R4] focuses on randomized smoothing in [R21], the leveraged methods are exactly the same. Specifically, adding Gaussian noise to the sample and averaging the adversarial perturbations, both used in randomized smoothing [R4] and in your proposed +G, have the exact same underlying mechanism. Additionally, the model used in [R4] is also a standardly trained model, which is still the same as your setting. Therefore, it is difficult to argue that your study presents a different scenario. The claim that adding Gaussian noise and then averaging over adversarial perturbations on standardly trained models can derive robust features has already been discovered.
>
> Please note that references starting with R are referred to as the reviewer's references.
>
> The perturbations in [R4] are derived from a robust model, as the authors indicate in their abstract: "In this paper, we show that these perceptually aligned gradients (human-identifiable features) also occur under randomized smoothing, an alternative means of constructing adversarially-robust classifiers." The authors observe that as the model's robustness increases (increase the standard deviation of incorporated Gaussian noise), the human-identifiable features in the perturbations become pronounced. They also reported that human-identifiable features are not observed when noise is not incorporated. Consequently, they suggest that human-identifiable features are a consequence of a robust model.
>
> In contrast, our study reveals that human-identifiable (robust) features are inherently present in standard models. By averaging perturbations from different standard models, even those without Gaussian noise, we uncover human-identifiable features. Notably, this averaging process does not introduce any new information, implying that these robust features are originally embedded in perturbations from standard models. This finding is very different from the conclusions drawn in [R4].
>
> Apparently, our method and conclusion differ from [R4].
>
>
> >Still, I maintain my viewpoint that ensembling multiple models into one can improve robustness, albeit not significantly, and thus improve perceptibility. First, it has been demonstrated that model ensemble can enhance adversarial robustness [R22-R23], thus leading to robust features. While the mentioned works focus on adversarially trained models, their assertion about the connection between ensembling and adversarial robustness can be easily extended to non-robust models. Furthermore, as stated in [R5], the emergence of perceptual aligned gradients (for a single image, without being perturbed by noise) exhibits a strong correlation with model robustness. Therefore, since both methods of averaging over models and introducing perturbations contribute to improving robustness, recognized as randomized smoothing and model ensemble, these conclusions are undoubtedly anticipated.
>
> [R5] does not make the claim that “the emergence of perceptual aligned gradients exhibits a strong correlation with model robustness.” Instead in the abstract of [R5], it states “We perform experiments to show that interpretable and perceptually aligned gradients are present even in models that do not show high robustness to adversarial attacks.”. **This statement differs significantly from the reviewer’s assertion** and, on the contrary, implicitly suggests that perceptually aligned gradients may not necessarily be related to the model’s robustness.
>
> There is no indication in the papers provided by the reviewer regarding the reviewer’s viewpoint that a slight increase in the model’s robustness will necessarily improve perturbations’ perceptibility. Furthermore, it is noteworthy that several defensive algorithms, such as defensive distillation, marginally enhance the model's robustness. We would be interested to know if the reviewer could offer thoughts on whether perturbations in these defensively augmented models exhibit a perception-aligned gradient.

---

> ### Author Response · Authors · 2023-11-23
> **Disagree With the Reviewer's Response (Part 2-2)**
>
> >The intuition here is that SGM [R7] modifies the algorithm for generating adversarial examples based on the design of the architecture, which is specifically tailored for residual connection models. From this, it can be inferred that different model architectures exhibit different non-robust features. **Intuitively, averaging various models leads to convergence towards robust features**.
>
> A key contribution of our paper is demonstrating the existence of robust features in perturbations derived from standard models, a finding previously unrecognized in the community, as highlighted by the reviewer's previous comment. In a previous comment, the reviewer noted "This paper challenges a well-acknowledged phenomenon in the context of adversarial robustness: the perceptual aligned gradient (PAG), which refers to the human-identifiable features that align with human perception in adversarial perturbations, **only** exists in robust models [1-3]." The reviewer also noted “There exist several works [8-10] that aim to explain the reason PAG **only** exists in robust models by characterizing the decision boundaries between different models, which is well supported by theoretical analysis.”
>
> In the previous comment, the reviewer stated: "In addition, it has also been shown [R7] that the distribution of non-robust features [R17] varies across different model architectures." As far as we are aware, this does not appear to have been addressed in [R7].
>
> >It seems that you missed Weakness 4. I would like to further discuss with you after you fulfill.
>
> We have conducted experiments and the results lead to our conclusion. It would be nice to build a theoretical model on what we have found, which is not done as confined by limited manpower and resources.

---

### Official Review · Reviewer_3kT3 · 2023-11-01

**Soundness:** 3 good
**Presentation:** 2 fair
**Contribution:** 2 fair
**Rating:** 5
**Confidence:** 4

**Summary:**

This paper delves into the exploration of the underlying reasons for adversarial perturbations. Specifically, the authors hypothesize that human-identifiable features are present within the perturbations, forming part of the inherent properties of these perturbations. To validate this hypothesis, the authors average perturbations generated by various neural networks to uncover the human-identifiable features.

**Strengths:**

+ This work finds that perturbations generated by existing methods statistically contain some human-identifiable features. These are clearly illustrated in the provided qualitative results.

+ To uncover these human-identifiable features, the authors use a simple method which averages extensive generated perturbations, which is reasonable.

+ This paper demonstrates that perturbations produced by certain attack methods converge at the object region.

+ This paper provides a clear narrative, supplemented by analytical insights.

**Weaknesses:**

- In the first paragraph of Section 4, on what basis do you assert that (1) the noise in perturbations is independent and (2) two perturbations from different models display distinct human-identifiable features? I couldn't find any references or evidence supporting the claims.

- The gradient-based attacks, proposed five years ago, aren't sufficiently contemporary to test the paper's hypothesis. There exist many newer gradient-based attacks, such as [1, 2].

- I observed that detecting human-identifiable features necessitates 2,700 samples (270 models and 10 noise-infused seed samples). These may suggest that the averaged perturbation, generated by the three attacking methods, gravitates towards the object region. However, they don't confirm that in every model, the generated perturbations house human-identifiable features. Hence, a deeper experimental analysis regarding model selection and the integration of Gaussian noise would be beneficial, perhaps including more ablation studies (like MM, MM+G, SM+G).

- Why choose only 20 fixed classes out of 1,000? And a mere 200 samples seem insufficient to substantiate the claims made in the paper

- It's noted that perturbations of identical images from varying attack algorithms are presumably alike. However, the results don't include background noise similarity or image perturbation similarity. Providing experimental evidence for this would enhance the argument.

- The experimental analysis concerning the two distinct types of human-identifiable features (masking effect and generation effect) appears limited. Visualizing the perturbation for targeted attacks would be beneficial.

-  Does the visual perturbation come from cases where the attack was successful? How does the perturbation behave in the case of an unsuccessful attack?

- While the paper asserts findings across three different datasets, I could only locate a detailed attack accuracy comparison for ImageNet in Appendix E Table 1. It is not clear why the NOISE performance surpass that of IMAGE?

[1] Rony J, Hafemann L G, Oliveira L S, et al. Decoupling direction and norm for efficient gradient-based l2 adversarial attacks and defenses. ICCV 2019

[2] Wang X, He K. Enhancing the transferability of adversarial attacks through variance tuning. CVPR 2021.

**Questions:**

See the questions in the weaknesses.

---

> ### Author Response · Authors · 2023-11-22
> **Response to Reviewer 3kT3 (Part 1)**
>
> We thank the reviewer for providing valuable feedback on our work. In the following, we will individually respond to each question from the reviewer.
> >1. In the first paragraph of Section 4, on what basis do you assert that (1) the noise in perturbations is independent and (2) two perturbations from different models display distinct human-identifiable features? I couldn't find any references or evidence supporting the claims.
>
>
> (1) and (2) are assumptions based on existing literature or empirical observations, which inspired us to uncover human-identifiable features in perturbations through the process of averaging.
>
> For assumption (1), the existing literature suggests that the noise observed in gradients relative to the loss function might be meaningless local variations in partial derivatives [1]. Therefore, it is reasonable to assume that the local variations of gradients trained independently from different models should not be aligned or dependent on each other.
>
> Regarding assumption (2), our empirical observations show that two robust models can generate distinct human-identifiable features for the same image using the same attack algorithm. This leads to our hypothesis that perturbations from standard-trained models also contain distinct human-identifiable features.
>
> [1] Smoothgrad: Removing noise by adding noise. Daniel Smilkov, et al. Workshop on Visualization for Deep Learning. 2017.
>
> >2. I observed that detecting human-identifiable features necessitates 2,700 samples (270 models and 10 noise-infused seed samples). These may suggest that the averaged perturbation, generated by the three attacking methods, gravitates towards the object region. However, they don't confirm that in every model, the generated perturbations house human-identifiable features. Hence, a deeper experimental analysis regarding model selection and the integration of Gaussian noise would be beneficial, perhaps including more ablation studies (like MM, MM+G, SM+G).
>
> Thank you for your comment. We would like to note that we have included the MM setting in our ablation study, as detailed in Appendix A1. The results consistently align with those from the MM+G setting and demonstrate human-identifiable features.
>
> Based on your suggestion, we have conducted additional experiments to explore the contribution of each model towards the human-identifiable feature. This involved varying the number of models used to average perturbations and comparing their mean square error to the perturbations from the MM setting. We found that, to achieve Mean Square Error (MSE) convergence within 0.05, an average of 25 models are needed for the three attack algorithms. For a MSE of 0.02, 90 models are required, while an MSE of 0.01 necessitates 157 models, for detailed information, please refer to Appendix I.
>
>
> >3. Why choose only 20 fixed classes out of 1,000? And a mere 200 samples seem insufficient to substantiate the claims made in the paper.
>
> We have carefully selected 20 classes from the ImageNet validation set to maximize the diversity of our study, including classes on animals, plants, architecture, toys, transportation, utility, etc. We chose the first 10 images, instead of hand-picking, in each class to eliminate the possibility of human bias. Additionally, we tested the results across three different datasets and with up to five attack algorithms. We consistently observed the emergence of human-identifiable features from perturbations in each and every case.
>
> While we are eager to expand our experimental scope, the computational costs are prohibitively high. For instance, the DeepFool attack in our experiments alone required over 250 hours on a Tesla V100 GPU. Thus, further expanding the experiment's scale is challenging. However, we are confident that the consistent results obtained across various settings are sufficient to support our argument.

---

> > ### Author Response · Authors · 2023-11-22
> > **Response to Reviewer 3kT3 (Part 2)**
> >
> > >4. It's noted that perturbations of identical images from varying attack algorithms are presumably alike. However, the results don't include background noise similarity or image perturbation similarity. Providing experimental evidence for this would enhance the argument.
> >
> > The reason why the cosine similarity will converge regardless of attack algorithms is because the masking effect reduces contrasts on human-identifiable features in an image and those features should not be dependent on the attack algorithm used. In section 5.2.3, we discuss why human-identifiable features are primarily located in the contour part of the perturbations. Therefore, computing the cosine similarity for the background part may not align with our objective of assessing the similarity of the masking effect across different attack algorithms. Since the background contains significantly fewer human-identifiable features, its inclusion is expected to reduce the overall cosine similarity score.
> >
> > The table below presents cosine similarity scores between perturbations generated by different attack algorithms under the MM+G setting. The first row indicates which two attack algorithms are used for generating perturbations, while the second and third rows compare the cosine similarity between the entire perturbations and their contour parts, respectively.
> >
> > | Attack Algorithms | BIM/CW | BIM/DeepFool | CW/DeepFool |
> > |-------------------|--------|--------------|-------------|
> > | Whole             | 0.35   | 0.41         | 0.59        |
> > | Contour           | 0.43   | 0.47         | 0.64        |
> >
> >
> >
> >
> > >5. The experimental analysis concerning the two distinct types of human-identifiable features (masking effect and generation effect) appears limited. Visualizing the perturbation for targeted attacks would be beneficial.
> >
> > We have presented the results for targeted attacks in Section 6, which serve to demonstrate the generation effect. In Appendix H, more targeted attack examples are included.
> >
> >
> > >6. Does the visual perturbation come from cases where the attack was successful? How does the perturbation behave in the case of an unsuccessful attack?
> >
> > The experiment is conducted under black-box attacks. Therefore, the attack's success depends on the testing model used. In the human evaluation test, we used all perturbations generated from the experiment, including those that failed to fool any of the testing models.
> >
> > For the BIM attack, we found that 14 perturbations, containing human-identifiable features under the MM+G setting, could lower the label class score. However, none of the 4 testing models used in the paper were fooled by any of these perturbations. This is attributed to the fact that the testing models exhibit high confidence scores for the respective images, making them more challenging to deceive.
> >
> > For those 14 images, the confidence score, derived from predictions processed via the SoftMax function, was, on average, 11.92% higher than those for images correctly classified in the experiment. This comparison is detailed in the following table, where the first column denotes the respective testing models. The table entries consist of the confidence scores for the 14 images that were not deceived by the perturbations, as well as the average confidence scores for all correctly classified images across the four testing models.
> >
> > We further discovered that when the $L_{inf}$ norm of the perturbations was increased to 0.04, only four images incorporated with perturbations remained correctly classified by all 4 testing models. This finding suggests that these perturbations, while initially unable to fool models due to high confidence scores, can still fool testing models as $L_{inf}$ norm is increased.
> >
> > | Model       | ResNet-50 | BN-Inception | DenseNet-121 | VGG-16 |
> > |-------------|-----------|--------------|--------------|--------|
> > | 14 Images   | 93.15%    | 88.29%       | 88.86%       | 79.33% |
> > | Avg.        | 82.54%    | 80.46%       | 78.32%       | 71.08% |
> >
> >
> > >7. While the paper asserts findings across three different datasets, I could only locate a detailed attack accuracy comparison for ImageNet in Appendix E Table 1.
> >
> > We would like to remind the reviewer that the attack accuracy for the remaining two datasets is located in Appendix C.3.

---

### Official Review · Reviewer_G5Xu · 2023-11-09

**Soundness:** 2 fair
**Presentation:** 3 good
**Contribution:** 1 poor
**Rating:** 5
**Confidence:** 3

**Summary:**

This paper conducted interesting analysis on the human-identifiable features concealed in adversarial perturbations crafted by different attack algorithms. In order to obtain the visual-recognizable patterns from gradient-driven adversarial perturbations, multi-samplings on different threat models was used based on the independence assumption. In experiment sections, thsi paper conducted such analysis on various threat models (274 in total) with various attack algorithms (gradient-based, search-based), which is efficient and solid. While the resulting denoised adversarial perturbations seem to have some clear pattern which can be recognized by human, the pure adversarial perturbation cannot reveal any information regarding the image itself. This paper also contains following discussion on the denoised adv perturbation by quantitatively analyzing its recognizability, checking its attack strength, and applying contour extraction. The overall analysis is plentiful and the results looks interesting.

**Strengths:**

- Evaluation on a large amount of threat models and attack algorithms make the whole experimental results to be reliable.
- Motivation on exploring the human-identifiable features directly, instead of applying XAI methods to interpret, looks efficient and interesting.
- Overall written is clear and easy to follow.

**Weaknesses:**

While I do appreciate such important and intense work on exploring the explainability in adversarial perturbations, I still have some major concerns about the whole paper.

- Human-identifiable features looks vague: I still remain unclear about how to logically define the "human-identifiable" here: In section 5.2.1 authors conducted recognizability experiments on these denoised adv perturbations but it can only prove they are "model-identifiable". We cannot make such claim by showing part of (or even all) extracted adv perturbations and they are all human-identifiable. Some human-labeling experiments is required as a strong evidence to prove this.

- The overall finding is not surprising: while it is good to see that denoised adversarial perturbation is similar to its corresponding raw image, I'm not surprising to see because gradient-based attacks perturb models' prediction by optimizing the objective function following the pixel-gradient direction --- larger pixel gradients indicate pixels here are important for threat model to identify this input image. Thus the outcome of gradient optimization, adversarial perturbation, should contain some important features to identify this image. And for search-based attacks, it still tend to follow the important pixels to craft their perturbation. I think this paper should focus more on the target-attack scenario - so we have our raw-image key features and our targeted label --- how would the adversarial perturbation be to reflect both concept? Currently it only has a very short paragraph discussing such scenario (Section 6).

**Questions:**

I put all my concerns to the weakness part and I do think this paper has a lot of space to improve.

However, I think the overall results is plentiful and interesting for other researchers to know (especially on denoised perturbation under targeted attack scenario). It could be a very interesting workshop paper after reorganizing it into a logical way.


======================================================

Updates after reading authors' rebuttal:

I really appreciate authors efforts on further elaborating the importance of their findings - now I tend to believe this is an interesting finding to me and it could inspire several future papers for further theoretical analysis. However, after checking Reviewer ZsSi's comments, there could be some literatures implicitly discussing such scenario but this paper lacks contribution on further exploring the underlying reasons. I would like to raise my score to 5 but reduce my confidence to 3.

---

> ### Author Response · Authors · 2023-11-22
> **Thank You for Your Comment**
>
> First, we would like to thank the reviewer for their detailed review of the paper. Please find our response below.
>
> >1. Human-identifiable features looks vague: I still remain unclear about how to logically define the "human-identifiable" here: In section 5.2.1 authors conducted recognizability experiments on these denoised adv perturbations but it can only prove they are "model-identifiable". We cannot make such claim by showing part of (or even all) extracted adv perturbations and they are all human-identifiable. Some human-labeling experiments is required as a strong evidence to prove this.
>
> We would like to note that we have conducted a human evaluation test on perturbations generated under the MM+G setting, as detailed in Section 5.2.1. An average accuracy of 80.7% is achieved by human participants, indicating that adversarial perturbations are highly classifiable/identifiable by humans.
>
> >2. The overall finding is not surprising: while it is good to see that denoised adversarial perturbation is similar to its corresponding raw image, I'm not surprising to see because gradient-based attacks perturb models' prediction by optimizing the objective function following the pixel-gradient direction --- larger pixel gradients indicate pixels here are important for threat model to identify this input image. Thus the outcome of gradient optimization, adversarial perturbation, should contain some important features to identify this image. And for search-based attacks, it still tend to follow the important pixels to craft their perturbation. I think this paper should focus more on the target-attack scenario - so we have our raw-image key features and our targeted label --- how would the adversarial perturbation be to reflect both concept? Currently it only has a very short paragraph discussing such scenario (Section 6).
>
> We thank the reviewer for providing us with the chance to clarify our work. In the following discussion, we will introduce the prevailing perspective on adversarial perturbations, and then explain the contribution of our work.
>
> Although adversarial perturbations are expected to contain features beneficial to identifying images, there is no guarantee that these features will be human-understandable. In fact, the prevailing view in the field is that human and machine rely on different types of features for classification. Adversarial perturbations, as a direct consequence of human-AI misalignment, are a type of feature (non-robust feature) that is highly informative for models yet incomprehensible to humans [1]. There is even a stronger version of the assumption stating that human-identifiable (robust) features should only exist in robust models, as suggested by reviewer 4 (ZsSi).
>
> The reviewer might think that, with the invention of explainable AI tools, it should appear straightforward that gradient/adversarial perturbations must include features identifiable by human. However, these tools often require preprocessing that largely alters the image, which could mask the actual features relied upon by the model. Consequently, the reported results may be ad-hoc and not accurately reflect reality [2].
>
> Our research reveals that human-identifiable features inherently reside in perturbations from a standardly trained model, offering fresh insights into explaining the properties of perturbations including transferability, the relation between adversarial training and increased model explainability, and the human-AI misalignment experiment, all of which are crucial in understanding the mechanisms behind adversarial perturbations.
>
> References:
>
> [1] Adversarial Examples Are Not Bugs, They Are Features. Andrew Ilyas, et al. NeurIPS, 2019.
>
> [2] Adversarial Examples and Human-ML Alignment. Alexender Madry. MIT CBMM Talks (https://www.youtube.com/watch?v=AvcRBuFreFg, 22:52-23:13)

---

### Official Review · Reviewer_QGNn · 2023-11-09

**Soundness:** 3 good
**Presentation:** 2 fair
**Contribution:** 2 fair
**Rating:** 5
**Confidence:** 5

**Summary:**

This paper studies how to extract human-identifiable features from adversarial examples. Based on the fact that DNN models are trained on human-labeled datasets, the authors assume that adversarial perturbations should also contain human-identifiable features.

The authors first claify that two factors, excessive gradient noise and incomplete features, hinder feature extraction. Therefore, the authors propose to utilize noise augmentations and model ensembling to mitigate these negative effects. The authors find two interesting phenomenons: masking effect (untargeted attacks) and generation effect (targeted attacks).

**Strengths:**

1. This problem is interesting. I like this topic.

2. The visualization results are also promising.

**Weaknesses:**

1. Although this problem is interesting, the authors do not provide more surprising findings and insights compared with previous works.
  1.1 Adversarial perturbations contain meanful or human-identifiable features have been studied in these works [1,2]. They may correspond to "robust" features.
  1.2 The proposed methods, noise augmentations and model ensembling are widely used in transfer attacks. More transfeable perturbations contain more "robust" features (human-identifiable features) and share more non-robust features. The previous work have shown this point [1]
  1.3 Although the visualizations are very promising, we are uncertain about the extent of assistance this can provide.

2. Some claims in the article are unclear:
  2.1 The two obtacles are not very clear. The first one (noisy gradient) is easy to understand. Lots of transfer attacks also propose to mitigate this negative effect to improve adversarial tranferability. However, there is insufficient evidence to support the second claim about incomplete learned features. Could you please provide more details about the second one?
  2.2 Meanwhile, the comparison between these two points is also unclear. Which factor has a greater negative impact on extracting human-identifiable features? As shown in experimental setting, the authors need to use lots of ensembling models. This has made is method less practical.
  2.3 The findings from Section 5.2.3 are interesting. The authors use the contour features to attack models. It also shows that contour features are important than background information. Could the authors please discuss connections and differences between this phenomenon and this work [3]?

3. Could the authors please provide more results about generation effect on targeted attacks?

[1]. Adversarial Examples Are Not Bugs, They Are Features.
[2]. Image Synthesis with a Single (Robust) Classifier.
[3] ImageNet-trained CNNs are biased towards texture: increasing shape bias improves accuracy and robustness

**Questions:**

Please see Weaknesses part.

---

> ### Author Response · Authors · 2023-11-22
> **Response to Reviewer QGNn (Part 1)**
>
> We thank the reviewer for providing valuable feedback on our work. In the following, we will individually respond to each question from the reviewer.
>
> >1. Although this problem is interesting, the authors do not provide more surprising findings and insights compared with previous works.\
> 1.1 Adversarial perturbations contain meanful or human-identifiable features have been studied in these works [1,2]. They may correspond to "robust" features.\
> 1.2 The proposed methods, noise augmentations and model ensembling are widely used in transfer attacks. More transfeable perturbations contain more "robust" features (human-identifiable features) and share more non-robust features. The previous work have shown this point [1].\
> 1.3 Although the visualizations are very promising, we are uncertain about the extent of assistance this can provide.
> >
>
> Note: References cited by the reviewer are cited as R#. For example, the first reference cited by the
> reviewer is indicated as R1.
>
> We thank the reviewer for giving us the chance to clarify our contributions. Our contribution is not to re-discover robust (human-identifiable) features but to demonstrate that robust features inherently exist in adversarial perturbations generated from a standard-trained model. In fact, the prevailing view in this field is that human and machine rely on different types of features for classification. This view leads to the claim that adversarial perturbations from standard models, as a direct consequence of human-AI misalignment, are a type of feature (non-robust feature) that is highly informative for models yet incomprehensible to human [R1].
>
>
> With our experiment, we demonstrate that perturbations from standard models inherently contain robust features, instead of mere non-robust features. To our best knowledge, this finding is yet to be discussed in the literature and provides a new perspective on understanding the mechanism governing the attack of perturbations on models.
>
> Reference [R2] illustrates the effectiveness of robust models in image generation and restoration tasks. A related study [1] by the same research group delves into the reasons behind this, which is closely related to our work. The study shows that gradients and perturbations generated by robust models align more closely with human perception, while perturbations from standard networks are significantly noisier and may not be human-identifiable. According to the authors, standard models learn different features than robust models.
>
> We show that, by averaging perturbations from standard models, robust features are still observable, indicating a shared learning aspect between standard and robust models. Furthermore, we offer a new yet plausible explanation for the increased perceptibility in adversarial training: Adversarial training could act as gradient regularization, reducing noise and homogenizing gradients, thus making perturbations more perceptible to humans, as stated in Section 7.2.
>
> Our study uncovered human-identifiable (robust) features in perturbations from standard training, shedding new light on perturbations’ properties, such as transferability, the link between adversarial training and enhanced model explainability, and the human-AI misalignment experiment. These aspects are key to understanding the mechanism of adversarial perturbations, allowing us to pursue adversarial-perturbations related applications.
>
> Reference:
>
> [1] Robustness May Be at Odds with Accuracy. Dimitris Tsipras, et al. ICLR 2019.

---

> ### Author Response · Authors · 2023-11-22
> **Response to Reviewer QGNn (Part ２)**
>
> >2. Some claims in the article are unclear:\
> 2.1 The two obtacles are not very clear. The first one (noisy gradient) is easy to understand. Lots of transfer attacks also propose to mitigate this negative effect to improve adversarial tranferability. However, there is insufficient evidence to support the second claim about incomplete learned features. Could you please provide more details about the second one? \
> 2.2 Meanwhile, the comparison between these two points is also unclear. Which factor has a greater negative impact on extracting human-identifiable features? As shown in experimental setting, the authors need to use lots of ensembling models. This has made is method less practical.\
> 2.3 The findings from Section 5.2.3 are interesting. The authors use the contour features to attack models. It also shows that contour features are important than background information. Could the authors please discuss connections and differences between this phenomenon and this work [3]?
> >
>
> **2.1** The obstacles hindering the presence of robust features are assumptions made in the paper based on existing literature or empirical observations. These obstacles motivate us to propose the method of averaging perturbations as a method to uncover these robust features.
>
> Regarding assumption (2), our empirical observations indicate that gradient/perturbations from robust models only contain a subset of human-identifiable features. This leads to the hypothesis that robust features from standard models should also contain incomplete human-identifiable features.
>
> We believe the reason behind is due to the capacity limit and challenge in learning optimum, neural networks may not completely utilize all robust feature for classification. Perturbations, designed to maximize the model's loss function, impact only those features the model recognizes. Consequently, these perturbations might encompass only a subset of features identifiable by humans.
>
>
> **2.2** Our research aims to investigate robust features in perturbations from standard models. Assessing which factors, noise or incompleteness, more significantly hinder robust features is not our primary focus. We recognize the value of such evaluations.  However, discerning the dominant factor is challenging due to the concurrent presence of noise and the incompleteness of robust features in perturbations. Separating one factor without altering the perturbations' nature may be challenging. This issue deserves our attention for further investigation.
>
> **2.3** In [R3], the authors demonstrates that Convolutional Neural Networks (CNNs) is used to classify images using both texture and contour, with texture playing a more significant role than in human classification, which relies more on contour.
>
> We segment the perturbations into those within the contour of the object and those in the background. We find that the contour parts contain both the object’s edges and textures that are similar to the original image. Figure 2 illustrates those perturbations from MM+G display texture-like features. For instance, one can observe a green, granular texture on the frog's back and a stripe-like texture on the baseball. These observations align with the findings in [R3] because, to deceive the model, perturbations may need to reduce the pixel values of both texture and contour features, which serve as critical factors influencing the model's decision.
>
> >3. Could the authors please provide more results about generation effect on targeted attacks?
>
> **3** Additional results about targeted attack have been provided in Appendix H. Thank you for your suggestion.

---

### Author Response · Authors · 2023-11-23
**Clarification of Our Contributions**

First, we would like to express our gratitude to all reviewers for the time and effort spent on reviewing our paper. In light of a second comment from a reviewer who has a strong opinion against our work, we believe it is important to further clarify the contributions of our research.



**Community perspective**:

According to a prevailing perspective in the community, adversarial perturbations derived from standard models are non-robust (non-human-identifiable) features, unlike those derived from robust models, which exhibit robust (human-identifiable) features [1]. A key factor behind this discrepancy is the fundamentally different features learned by robust and standard models [2,3]. Building on this premise, some researchers argue that robust features should not be present in perturbations derived from standard models [2,4].

**Our findings**:

In our research, we demonstrate that perturbations derived from standard models inherently possess robust features. This is achieved by generating perturbations from different standard models and then averaging them.

Although the integration of ensembling methods and augmentation techniques has been shown to yield robust features in perturbations [5,6], researchers attribute this phenomenon to the increased robustness of the model brought about by these techniques [7].

Our averaging method eliminates such concerns, allowing us to state that perturbations derived from the standard model inherently include robust features. The reason for this is that averaging will not introduce new features to perturbations or alter their nature. This finding, to the best of our knowledge, is entirely new.



The discovery that adversarial perturbations from a standard model include human-identifiable (robust) features has led us to uncover two key properties of perturbations from the standard model:

1. A convergence phenomenon among perturbations generated using different attack algorithms.

2. An apparent stronger attack strength of the perturbation contour when compared with the background.

Furthermore, the presence of robust features in perturbations provides a straightforward explanation for several well-established properties of adversarial perturbations, including:

1. Transferability across different models.

2. Enhanced perceptibility during adversarial training.

3. Human-AI misalignment experiments [1].

We thank the reviewers for their time and sincerely hope this discovery can be seen by other researchers in the community.

**References**:\
[1] Adversarial Examples Are Not Bugs, They Are Features. Andrew Ilyas, et al. NeurIPS, 2019.\
[2] Robustness May Be at Odds with Accuracy. Dimitris Tsipras, et al. ICLR 2019.\
[3] Image Synthesis with a Single (Robust) Classifier. Shibani Santurkar, et al. NeurIPS 2019.\
[4] On the Connection Between Adversarial Robustness and Saliency Map Interpretability. Christian Etmann, et al. ICML 2019.\
[5] Adversarial Patch. Tom Brown, et al. arXiv 2017.\
[6] Synthesizing Robust Adversarial Examples. Anish Athalye, et al. CoRR 2017.\
[7] Are Perceptually-Aligned Gradients a General Property of Robust Classifiers? Christian Etmann, et al. NeurIPS 2019 Workshop.

---

### Meta-Review · Area_Chair_FNuU · 2023-12-12

**Metareview:**

This paper investigates human-identifiable features in adversarial perturbations. Gaussian noise is used to identify these features. In targeted attacks, these features generate features or objects of the target class. In untargeted attacks, these features hide features or objects of the original class. This phenomenon can explain some properties of adversarial perturbations.

Strengths: This paper revisits the fundamental mechanisms of adversarial perturbations, a problem of importance. This paper presents human studies that support the hypothesis that the emergence of semantic features is not coincidental. The hypothesis is validated across targeted and untargeted attacks, and includes search-based attacks.

Weaknesses: Upon reviewing the discussions between Reviewer ZsSi and the authors, I agree with some of the concerns raised by Reviewer ZsSi, particularly with respect to the connections to prior work. I have also read reference [4], which was provided by Reviewer ZsSi. It appears to me that the main difference between [4] and this paper is the order of the maximization (for finding the perturbation) and the average (expectation in [4]). However, [4] and this paper still share a number of similarities. The authors may want to consider comparing their proposed method to [4] in the experiments. Reviewer ZsSi also mentions other connections to other prior works, which the authors should consider in future revisions.
In addition, I find that this paper is not well-written in terms of its mathematical notation. The meaning of Equation (1) is unclear. I would recommend using random variables rather than the symbol for the normal distribution in Equation (1). Additionally, the perturbation used in Equation (1) should be more formally defined, although it is possible to infer what it should be.

**Justification For Why Not Higher Score:**

See the weaknesses above.

**Justification For Why Not Lower Score:**

N/A

---

### Decision · Program_Chairs · 2024-01-16

Reject